# MaxInfoRL: Boosting exploration in reinforcement learning through information gain maximization

**Bhavya Sukhija**[*,1], **Stelian Coros**[1] **Andreas Krause**[1], **Pieter Abbeel**[2], **Carmelo Sferrazza**[2]
ETH Zürich [1], UC Berkeley[2]
{sukhijab, scoros, krausea}@ethz.ch
{pabbeel, csferrazza}@berkeley.edu

## Abstract

Reinforcement learning (RL) algorithms aim to balance exploiting the current best strategy with exploring new options that could lead to higher rewards. Most common RL algorithms use undirected exploration, i.e., select random sequences of actions. Exploration can also be directed using intrinsic rewards, such as curiosity or model epistemic uncertainty. However, effectively balancing task and intrinsic rewards is challenging and often task-dependent. In this work, we introduce a framework, MaxInfoRL, for balancing intrinsic and extrinsic exploration. MaxInfoRL steers exploration towards informative transitions, by maximizing intrinsic rewards such as the information gain about the underlying task. When combined with Boltzmann exploration, this approach naturally trades off maximization of the value function with that of the entropy over states, rewards, and actions. We show that our approach achieves sublinear regret in the simplified setting of multi-armed bandits. We then apply this general formulation to a variety of off-policy model-free RL methods for continuous state-action spaces, yielding novel algorithms that achieve superior performance across hard exploration problems and complex scenarios such as visual control tasks.

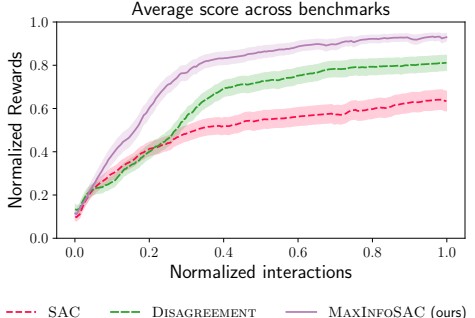

(a) Normalized average performance of MaxInfoRL across several deep RL benchmarks on state-based tasks.

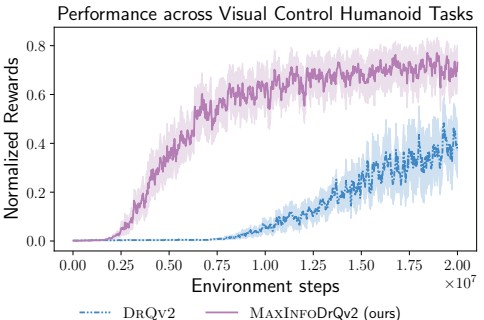

(b) Normalized average performance of MaxInfoDRQv2 on the humanoid visual control tasks (stand, walk, and run).

Figure 1: We summarize the normalized performance of different variants of MaxInfoRL; MaxInfoSAC for state-based control and MaxInfoDRQv2 for visual control (cf., Section 4 for more details). We report the mean performance across five seeds with one standard error.

## 1 Introduction

Reinforcement learning (RL) has found numerous applications in sequential decision-making problems, from games (Silver et al., 2017), robotics (Hwangbo et al., 2019; Brohan et al., 2023), to fine-tuning of large language models (Ouyang et al., 2022). However, most widely applied RL algorithms such as PPO (Schulman et al., 2017) are inherently sample-inefficient, requiring hundreds of hours of environment interactions for learning. Off-policy methods like SAC (Haarnoja et al., 2018), REDQ (Chen et al., 2021), and DroQ (Hiraoka et al., 2022) offer a more sample and compute efficient alternative and have demonstrated success in real-world learning (Smith et al.,

---

Open-source implementations: `https://sukhijab.github.io/projects/maxinforl/`

2022). Despite this, they often require dense reward signals and suffer in the presence of sparse rewards or local optima. This is primarily due to their use of naive exploration schemes such as $\epsilon$-greedy or Boltzmann exploration and effectively take random sequences of actions for exploration. These strategies are known to be suboptimal even for basic tasks (Cesa-Bianchi et al., 2017; Burda et al., 2018; Sukhija et al., 2024b), yet remain popular due to their simplicity and scalability.

Several works (Burda et al., 2018; Pathak et al., 2017; 2019; Sekar et al., 2020; Sukhija et al., 2024b) use intrinsic reward signals, e.g., curiosity or information gain, to improve the exploration of RL agents. Moreover, information gain (Lindley, 1956) is also widely applied in Bayesian experiment design (Chaloner & Verdinelli, 1995) and is the basis of many active learning methods (Krause et al., 2008; Balcan et al., 2010; Hanneke et al., 2014; Hübotter et al., 2024). In RL, exploration via maximizing information gain offers strong theoretical guarantees (Mania et al., 2020; Sukhija et al., 2024b) and achieves state-of-the-art empirical performance (Sekar et al., 2020; Mendonca et al., 2021). However, a significant gap persists in both the theoretical and practical understanding of how to effectively balance intrinsic exploration objectives with naive extrinsic exploration algorithms. The goal of this work is to bridge this gap. To this end, we revisit the traditional, widely-used Boltzmann exploration, and enhance it by incorporating exploration bonuses derived from intrinsic rewards, like information gain. Our approach is grounded in both theoretical insights and practical motivations, and we empirically validate it across several deep RL benchmarks.

The key contributions of this work are summarized as follows:

**Contributions**

1. We propose MAXINFORL, a novel class of off-policy model-free algorithms for continuous state-action spaces that augments existing RL methods with directed exploration. In essence, MAXINFORL builds on standard Boltzmann exploration and guides it via an intrinsic reward. We propose a practical auto-tuning procedure that largely simplifies trading off extrinsic and intrinsic objectives. This yields algorithms that explore by visiting trajectories that achieve the maximum information gain about the underlying MDP, while efficiently solving the task. As a result, MAXINFORL retains the simplicity of traditional RL methods while adding directed exploration through intrinsic rewards. Additionally, we show how the same idea can be combined with other naive exploration techniques, such as $\epsilon$–greedy.

2. In the simplified setting of stochastic multi-armed bandits in continuous spaces, we show that MAXINFORL has sublinear regret. In addition, we show that MAXINFORL benefits from all theoretical properties of contraction and convergence that hold for max-entropy RL algorithms such as SAC (Haarnoja et al., 2018).

3. In our experiments, we use an ensemble of dynamics models to estimate information gain and combine MAXINFORL with SAC (Haarnoja et al., 2018), REDQ (Chen et al., 2021), DrQ (Yarats et al., 2021), and DrQv2 (Yarats et al., 2022). We evaluate it on standard deep RL benchmarks for state and visual control tasks and show that MAXINFORL performs the best across all tasks and baselines, obtaining the highest performance also in challenging exploration problems (see., Fig. 1 for the average performance of MAXINFORL across several environments).

## 2 BACKGROUND

A core challenge in RL is deciding whether the agent should leverage its current knowledge to maximize rewards or try new actions in pursuit of better solutions. Striking this balance between exploration–exploitation is critical. Here, we first introduce the problem setting, then we discuss two of the most commonly used exploration strategies in RL: $\epsilon$-greedy and Boltzmann exploration.

### 2.1 PROBLEM SETTING

We study an infinite-horizon Markov decision process (MDP, Puterman, 2014), defined by the tuple $(\mathcal{S}, \mathcal{A}, p, \gamma, r, \rho)$, where the state and action spaces are continuous, i.e., $\mathcal{S} \subset \mathbb{R}^{d_s}, \mathcal{A} \subset \mathbb{R}^{d_a}$, and the unknown transition kernel $p : \mathcal{S} \times \mathcal{S} \times \mathcal{A} \to [0, \infty)$ represents the probability density of the next state $s_{t+1} \in \mathcal{S}$ given the current state $s_t \in \mathcal{S}$ and action $a_t \in \mathcal{A}$. At each step $t$ in the environment, the agent observes the state $s_t$, samples an action $a_t$ from the policy $\pi : \mathcal{A} \times \mathcal{S} \to [0, \infty), a_t \sim \pi(a|s_t)$, and receives a reward $r : \mathcal{S} \times \mathcal{S} \times \mathcal{A} \to [-\frac{1}{2}r_{\max}, \frac{1}{2}r_{\max}]$. The agent's goal is to learn a policy $\pi^*$ that maximizes the $\gamma$ discounted reward w.r.t. the initial state distribution $s_0 \sim \rho$.

$$\pi^* = \arg\max_{\pi \in \Pi} J(\pi) = \arg\max_{\pi \in \Pi} \mathbb{E}_{s_0, a_0, \dots} \left[ \sum_{t=0}^{\infty} \gamma^t r_t \right]. \tag{1}$$

In the following, we provide the definitions of the state-action value function $Q^{\pi}$ and the value function $V^{\pi}$:

$$Q^{\boldsymbol{\pi}}(\boldsymbol{s}_t, \boldsymbol{a}_t) = \mathbb{E}_{\boldsymbol{s}_{t+1}, \boldsymbol{a}_{t+1} \sim \boldsymbol{\pi}, \ldots} \left[ \sum_{l=0}^{\infty} \gamma^l r_{t+l} \right], \ V^{\boldsymbol{\pi}}(\boldsymbol{s}_t) = \mathbb{E}_{\boldsymbol{a}_t \sim \boldsymbol{\pi}, \boldsymbol{s}_{t+1}, \boldsymbol{a}_{t+1} \sim \boldsymbol{\pi}, \ldots} \left[ \sum_{l=0}^{\infty} \gamma^l r_{t+l} \right].$$

## 2.2 $\epsilon-$GREEDY AND EXPLORATION

The $\epsilon$-greedy strategy (Kearns & Singh, 2002; Mnih, 2013; Van Hasselt et al., 2016) is widely applied in RL to balance exploration and exploitation, where the RL agent follows this simple decision rule below to select actions

$$\boldsymbol{a}_t = \begin{cases} \boldsymbol{a} \sim \text{Unif}(\mathcal{A}) & \text{with probability } \epsilon_t \\ \arg\max_{\boldsymbol{a} \in \mathcal{A}} Q^*(\boldsymbol{s}_t, \boldsymbol{a}) & \text{else,} \end{cases} \tag{2}$$

Here $Q^*$ is the estimate of the optimal state-action value function. Therefore, at each step $t$, with probability $\epsilon_t$, a random action $\boldsymbol{a}_t \sim \text{Unif}(\mathcal{A})$ is sampled, else the greedy action $\boldsymbol{a}_t = \max_{\boldsymbol{a} \in \mathcal{A}} Q^*(\boldsymbol{s}_t, \boldsymbol{a})$ is picked. Lillicrap (2015); Fujimoto et al. (2018) extend this strategy to continuous state-action spaces, where a deterministic policy $\boldsymbol{\pi}_{\theta}$ is learned to maximize the value function and combined with random Gaussian noise for exploration.

## 2.3 BOLTZMANN EXPLORATION

Boltzmann exploration is the basis of many RL algorithms (Sutton, 2018; Szepesvári, 2022). The policy distribution $\boldsymbol{\pi}$ for Boltzmann is represented through

$$\boldsymbol{\pi}(\boldsymbol{a}|\boldsymbol{s}) \propto \exp\left(\alpha^{-1} Q^{\boldsymbol{\pi}}(\boldsymbol{s}, \boldsymbol{a})\right), \tag{3}$$

where $\alpha$ is the temperature parameter that regulates exploration and $Q^{\boldsymbol{\pi}}$ is the soft-$Q$ function. We neglect the normalization term $Z^{-1}(\boldsymbol{s})$ in the definition for simplicity. As $\alpha \to 0$, the policy greedily maximizes $Q^{\boldsymbol{\pi}}(\boldsymbol{s}, \boldsymbol{a})$, i.e. it exploits, and as $\alpha \to \infty$ the policy adds equal mass to all actions in $\mathcal{A}$, effectively performing uniform exploration. Intuitively, Boltzmann exploration can be interpreted as a smoother alternative to $\epsilon-$greedy, with $\alpha$ serving a similar role to $\epsilon$ in controlling the degree of exploration. Cesa-Bianchi et al. (2017) show that the standard Boltzmann exploration is suboptimal even in the simplest settings. They highlight that a key shortcoming of Boltzmann exploration is that it does not reason about the uncertainty of its estimates.

Overall, both $\epsilon-$greedy and Boltzmann exploration strategies are undirected. They fail to account for the agent's "lack of knowledge" and do not encourage risk- or knowledge-seeking behavior. The agent explores by sampling random action sequences, which leads to suboptimal performance, particularly in challenging exploration tasks with continuous state-action spaces.

## 2.4 INTRINSIC EXPLORATION WITH INFORMATION GAIN

Intrinsic rewards or motivation are used to direct agents toward underexplored regions of the MDP. Hence they enable RL agents to acquire information in a more principled manner as opposed to the aforementioned naive exploration methods. Effectively, the agent explores by selecting policies that maximize the $\gamma$-discounted intrinsic rewards. A common choice for the intrinsic reward is the information gain (Cover & Thomas, 2006; Sekar et al., 2020; Mendonca et al., 2021; Sukhija et al., 2024b). Accordingly, for the remainder of the paper, we center our derivations around using information gain as the intrinsic reward. However, our approach is flexible and can also be combined with other intrinsic exploration objectives, such as RND (Burda et al. (2018), see Appendix D).

We study a non-linear dynamical system of the form

$$\tilde{\boldsymbol{s}}_{t+1} = \boldsymbol{f}^*(\boldsymbol{s}_t, \boldsymbol{a}_t) + \boldsymbol{w}_t. \tag{4}$$

Here $\tilde{\boldsymbol{s}}_{t+1} = [\boldsymbol{s}_{t+1}^{\top}, r_t]^{\top}$ represents the next state and reward, $\boldsymbol{f}^*$ represents the *unknown dynamics* and reward function of the MDP and $\boldsymbol{w}_t$ is the process noise, which we assume to be zero-mean i.i.d., $\sigma^2$-Gaussian. Note this is a very common representation of nonlinear systems with continuous state-action spaces (Khalil, 2015) and the basis of many RL algorithms (Pathak et al., 2019; Kakade et al., 2020; Curi et al., 2020; Mania et al., 2020; Wagenmaker et al., 2023; Sukhija et al., 2024a). Furthermore, it models all essential and unknown components of the underlying MDP; the transition kernel and the reward function.

**Approximating information gain**  Given a dataset of transitions $\mathcal{D}_n = \{(\boldsymbol{s}_i, \boldsymbol{a}_i, \tilde{\boldsymbol{s}}_i')\}_{i=0}^n$, e.g., a replay buffer, we learn a Bayesian model of the unknown function $\boldsymbol{f}^*$, to obtain a posterior distribution $p(\boldsymbol{f}^*|\mathcal{D}_n)$ for $\boldsymbol{f}^*$. This distribution can be Gaussian, e.g., Gaussian process models (Rasmussen & Williams, 2005) or represented through Bayesian neural networks like probabilistic ensembles (Lakshminarayanan et al., 2017). As opposed to the typical model-based RL setting, similar to Burda et al. (2018); Pathak et al. (2017; 2019), our learned model is only used to determine the intrinsic reward. The information gain $I(\tilde{\boldsymbol{s}}'; \boldsymbol{f}^*|\boldsymbol{s}, \boldsymbol{a}, \mathcal{D}_n)$, reflects the uncertainty about the unknown dynamics $\boldsymbol{f}^*$ from observing the transition $(\boldsymbol{s}, \boldsymbol{a}, \tilde{\boldsymbol{s}}')$. Moreover, let $\boldsymbol{\sigma}(\boldsymbol{s}, \boldsymbol{a}|\mathcal{D}_n) = [\sigma_j(\boldsymbol{s}, \boldsymbol{a})]_{j \leq d_{\boldsymbol{s}}+1}$ denote the model epistemic uncertainty or disagreement of $\boldsymbol{f}^*$. Sukhija et al. (2024b, Lemma 1.) show that

$$I(\tilde{\boldsymbol{s}}'; \boldsymbol{f}^*|\boldsymbol{s}, \boldsymbol{a}, \mathcal{D}_n) = H(\tilde{\boldsymbol{s}}'|\boldsymbol{s}, \boldsymbol{a}, \mathcal{D}_n) - H(\tilde{\boldsymbol{s}}|\boldsymbol{s}, \boldsymbol{a}, \boldsymbol{f}^*, \mathcal{D}_n) \leq \underbrace{\sum_{j=1}^{d_{\boldsymbol{s}}+1} \log\left(1 + \frac{\sigma_{n-1,j}^2(\boldsymbol{s}_t, \boldsymbol{a}_t)}{\sigma^2}\right)}_{I_u(\boldsymbol{s}, \boldsymbol{a})} \quad (5)$$

where $H$ denotes the (differential) entropy (Cover & Thomas, 2006) and in Eq. (5) the equality holds when $p(\boldsymbol{f}^*|\mathcal{D}_n)$ is Gaussian. Note that while the above is an upper bound, Sukhija et al. (2024b) motivate this choice from a theoretical perspective proving convergence of the active learning algorithm for the model-based setting. In this work, similar to Pathak et al. (2019); Sekar et al. (2020); Mendonca et al. (2021); Sukhija et al. (2024b), we use the upper bound of the information gain for our practical algorithm. The upper bound has a natural interpretation, since by picking actions $\boldsymbol{a}_t$ that maximize it, we effectively visit areas where we have high uncertainty about the unknown function $\boldsymbol{f}^*$, therefore performing exploration in both state and action space. Empirically, this approach has shown to perform well, e.g., Sekar et al. (2020); Mendonca et al. (2021).

**Data dependence of intrinsic rewards**  Information gain and other intrinsic rewards depend on the data $\mathcal{D}_n$, making them inherently nonstationary and non-Markovian. Intuitively, underexplored areas of the MDP become less informative once visited (c.f., Prajapat et al. (2024) for more details). However, in RL, intrinsic rewards are often treated similarly to extrinsic rewards, a simplification that works very well in practice (Burda et al., 2018; Sekar et al., 2020). We take a similar approach in this paper and omit the dependence of $I$ on $\mathcal{D}_n$ and use $I(\tilde{\boldsymbol{s}}'; \boldsymbol{f}^*|\boldsymbol{s}, \boldsymbol{a})$ from hereon for simplicity.

## 3 MAXINFORL

In this section, we present our method for combining intrinsic exploration with classical exploration strategies. While MAXINFORL builds directly on Boltzmann exploration, we begin by illustrating its key ideas in the context of an $\epsilon$–greedy strategy, due to its mathematical simplicity and natural distinction between exploration and exploitation steps. The insights gained from this serve as motivation for developing our main method: MAXINFORL with Boltzmann exploration algorithms, which we evaluate in Section 4.

### 3.1 MODIFYING $\epsilon$–GREEDY FOR DIRECTED EXPLORATION

We modify the $\epsilon$–greedy strategy from Section 2.2 and learn two critics, $Q_{\text{extrinsic}}^*$ and $Q_{\text{intrinsic}}^*$, where $Q_{\text{extrinsic}}^*$ is the state-action value function of the extrinsic reward $r$ and $Q_{\text{intrinsic}}^*$ the critic of an intrinsic reward function $r_{\text{intrinsic}}$, for instance, the information gain (see Eq. (5)). Unlike traditional $\epsilon$–greedy exploration, we leverage intrinsic rewards to guide exploration more effectively by selecting actions that maximize $Q_{\text{intrinsic}}^*$, leading to more informed exploration rather than random sampling. At each step $t$, we pick a greedy action that maximizes $Q_{\text{extrinsic}}^*$ with probability $1 - \epsilon_t$, while for exploration, the action that maximizes the intrinsic critic is selected, i.e., $\boldsymbol{a}_t = \max_{\boldsymbol{a} \in \mathcal{A}} Q_{\text{intrinsic}}^*(\boldsymbol{s}_t, \boldsymbol{a})$.

$$\boldsymbol{a}_t = \begin{cases} \arg\max_{\boldsymbol{a} \in \mathcal{A}} Q_{\text{intrinsic}}^*(\boldsymbol{s}_t, \boldsymbol{a}) & \text{with probability } \epsilon_t \\ \arg\max_{\boldsymbol{a} \in \mathcal{A}} Q_{\text{extrinsic}}^*(\boldsymbol{s}_t, \boldsymbol{a}) & \text{else}, \end{cases} \quad (6)$$

We call the resulting exploration strategy $\epsilon$–MAXINFORL. This approach is motivated by the insight that in continuous spaces, intrinsic rewards cover the state-action spaces much more efficiently than undirected random exploration, making them more effective for exploration in general (Aubret et al., 2023; Sekar et al., 2020; Sukhija et al., 2024b). In Appendix A, to give a theoretical intuition of our approach, we study $\epsilon$–MAXINFORL in the simplified setting of multi-armed bandit (MAB). We show that as more episodes are played, it gets closer to the optimal solution, i.e. has sublinear-regret.

The key takeaway from $\epsilon$–MAXINFORL is that instead of exploring with actions that maximize entropy in the action spaces, e.g., uniform sampling, we select policies that also yield high information about the MDP during learning. In the following, we leverage this idea and modify the target distribution of Boltzmann exploration to incorporate intrinsic exploration bonuses. Moreover, $\epsilon$–MAXINFORL has two practical drawbacks; (*i*) it requires training two actor-critics and (*ii*) practically, the probability $\epsilon_t$ is specified by the problem designer. We address both these limitations in the section below and present our main method.

## 3.2 MAXINFORL WITH BOLTZMANN EXPLORATION

In Section 3.1, we modify $\epsilon$–greedy to sample actions with high intrinsic rewards during exploration instead of randomly picking actions. Motivated from the same principle, we augment the distribution of Boltzmann exploration with the intrinsic reward $I(\tilde{\boldsymbol{s}}'; \boldsymbol{f}^*|\boldsymbol{s}, \boldsymbol{a})$ to get the following

$$\boldsymbol{\pi}(\boldsymbol{a}|\boldsymbol{s}) \propto \exp\left(\alpha^{-1}Q^{\boldsymbol{\pi}}(\boldsymbol{s}, \boldsymbol{a}) + I(\tilde{\boldsymbol{s}}'; \boldsymbol{f}^*|\boldsymbol{s}, \boldsymbol{a})\right). \tag{7}$$

The resulting distribution encourages exploration w.r.t. information gain, with $\alpha$ playing a similar role to $\epsilon$ in Eq. (6). Therefore, Eq. (7) can be viewed as a *soft* formulation of Eq. (6). Effectively, instead of randomly sampling actions, for large values of the temperature, we pick actions that yield high information while maintaining the exploitative behavior for smaller temperatures. This distribution is closely related to the epistemic risk-seeking exponential utility function from $K$-learning (O'Donoghue, 2021) and probabilistic inference in RL (Tarbouriech et al., 2024). As we show in the following, this choice of parameterization results in a very intuitive objective for the policy. Given the previous policy $\boldsymbol{\pi}^{\text{old}}$ and $Q^{\boldsymbol{\pi}^{\text{old}}}$, akin to Haarnoja et al. (2018), we select the next policy $\boldsymbol{\pi}^{\text{new}}$ through the following optimization

$$\begin{aligned}
\boldsymbol{\pi}^{\text{new}} &= \arg\min_{\boldsymbol{\pi}\in\Pi} \mathrm{D}_{\mathrm{KL}}\left(\boldsymbol{\pi}(\cdot|\boldsymbol{s}) \middle\| Z^{-1}(\boldsymbol{s})\exp\left(\frac{1}{\alpha}Q^{\boldsymbol{\pi}^{\text{old}}}(\boldsymbol{s}, \cdot) + I(\tilde{\boldsymbol{s}}'; \boldsymbol{f}^*|\boldsymbol{s}, \boldsymbol{a})\right)\right) \\
&= \arg\max_{\boldsymbol{\pi}\in\Pi} \mathbb{E}_{\boldsymbol{a}\sim\boldsymbol{\pi}(\cdot|\boldsymbol{s})}\left[Q^{\boldsymbol{\pi}^{\text{old}}}(\boldsymbol{s}, \boldsymbol{a}) - \alpha\log(\boldsymbol{\pi}(\boldsymbol{a}|\boldsymbol{s})) + \alpha I(\tilde{\boldsymbol{s}}'; \boldsymbol{f}^*|\boldsymbol{s}, \boldsymbol{a})\right] \\
&= \arg\max_{\boldsymbol{\pi}\in\Pi} \mathbb{E}_{\boldsymbol{a}\sim\boldsymbol{\pi}(\cdot|\boldsymbol{s})}\left[Q^{\boldsymbol{\pi}^{\text{old}}}(\boldsymbol{s}, \boldsymbol{a})\right] + \alpha H(\tilde{\boldsymbol{s}}', \boldsymbol{a}|\boldsymbol{s}),
\end{aligned} \tag{8}$$

here in the last line we used that $\mathbb{E}_{\boldsymbol{a}\sim\boldsymbol{\pi}(\cdot|\boldsymbol{s})}[-\log(\boldsymbol{\pi}(\boldsymbol{a}|\boldsymbol{s})) + I(\tilde{\boldsymbol{s}}'; \boldsymbol{f}^*|\boldsymbol{s}, \boldsymbol{a})] = H(\boldsymbol{a}|\boldsymbol{s}) + H(\tilde{\boldsymbol{s}}'|\boldsymbol{a}, \boldsymbol{s}) - H(\tilde{\boldsymbol{s}}'|\boldsymbol{s}, \boldsymbol{a}, \boldsymbol{f}^*) = H(\tilde{\boldsymbol{s}}', \boldsymbol{a}|\boldsymbol{s}) - H(\boldsymbol{w})$. Hence, the policy $\boldsymbol{\pi}^{\text{new}}$ trades off maximizing the value function with the *entropy of the states, rewards, and actions*. This trade-off is regulated through the temperature parameter $\alpha$. We provide a different perspective to Eq. (8) from the lens of control as inference (Levine, 2018; Hafner et al., 2020) in Appendix C.

**Separating exploration bonuses** MAXINFORL has two exploration bonuses; (*i*) the policy entropy, and (*ii*) the information gain (Eq. (5)). The two terms are generally of different magnitude and tuning the temperature for the policy entropy is fairly well-studied in RL (Haarnoja et al., 2018). To this end, we modify Eq. (8) and introduce two individual temperature parameters $\alpha_1$ and $\alpha_2$ to separate the bonuses. Furthermore, since information gain does not have a closed-form solution in general, akin to prior work (Sekar et al., 2020; Sukhija et al., 2024b), we use its upper bound $I_u(\boldsymbol{s}, \boldsymbol{a})$ (Eq. (5)) instead.

$$\begin{aligned}
J^{\boldsymbol{\pi}^{\text{old}}}(\boldsymbol{\pi}|\boldsymbol{s}) &= \mathbb{E}_{\boldsymbol{a}\sim\boldsymbol{\pi}(\cdot|\boldsymbol{s})}\left[Q^{\boldsymbol{\pi}^{\text{old}}}(\boldsymbol{s}, \boldsymbol{a}) - \alpha_1\log(\boldsymbol{\pi}(\boldsymbol{a}|\boldsymbol{s})) + \alpha_2 I_u(\boldsymbol{s}, \boldsymbol{a})\right] \\
\boldsymbol{\pi}^{\text{new}}(\cdot|\boldsymbol{s}) &= \arg\max_{\boldsymbol{\pi}\in\Pi} J^{\boldsymbol{\pi}^{\text{old}}}(\boldsymbol{\pi}|\boldsymbol{s})
\end{aligned} \tag{9}$$

For $\alpha_1$, we can either use a deterministic policy with $\alpha_1 = 0$ like Lillicrap (2015) or auto-tune $\alpha_1$ as suggested by Haarnoja et al. (2018). Notably, for $\alpha_2 = 0$ we get the standard max entropy RL methods (Haarnoja et al., 2018). Therefore, by introducing two separate temperatures, we can treat information gain as another exploration bonus in addition to the policy entropy and combine it with any RL algorithm.

**Auto-tuning the temperature for the information gain bonus** Haarnoja et al. (2018) formulate the problem of soft-Q learning as a constrained optimization.

$$\begin{aligned}
\boldsymbol{\pi}^*(\cdot|\boldsymbol{s}) &:= \arg\max_{\boldsymbol{\pi}\in\Pi} \mathbb{E}_{\boldsymbol{a}\sim\boldsymbol{\pi}}\left[Q^{\boldsymbol{\pi}}(\boldsymbol{s}, \boldsymbol{a})\right] \text{ s.t., } H(\boldsymbol{a}|\boldsymbol{s}) \geq \bar{H} \\
&:= \arg\max_{\boldsymbol{\pi}\in\Pi} \min_{\alpha_1\geq 0} \mathbb{E}_{\boldsymbol{a}\sim\boldsymbol{\pi}}\left[Q^{\boldsymbol{\pi}}(\boldsymbol{s}, \boldsymbol{a}) - \alpha_1(\log(\boldsymbol{\pi}(\boldsymbol{a}|\boldsymbol{s})) + \bar{H})\right].
\end{aligned}$$

The entropy coefficient is then auto-tuned by solving this optimization problem gradually via stochastic gradient descent (SGD). In a similar spirit, we propose the following constraints to auto-tune the temperatures for the entropy and the information gain

$$\boldsymbol{\pi}^*(\cdot|\boldsymbol{s}) := \underset{\boldsymbol{\pi} \in \Pi}{\arg\max} \, \mathbb{E}_{\boldsymbol{a} \sim \boldsymbol{\pi}} \left[ Q^{\boldsymbol{\pi}}(\boldsymbol{s}, \boldsymbol{a}) \right] \text{ s.t., } H(\boldsymbol{a}|\boldsymbol{s}) \geq \bar{H}, \mathbb{E}_{\boldsymbol{a} \sim \boldsymbol{\pi}} \left[ I_u(\boldsymbol{s}, \boldsymbol{a}) \right] \geq \bar{I}_u(\boldsymbol{s}) \tag{10}$$

$$:= \underset{\boldsymbol{\pi} \in \Pi}{\arg\max} \, \underset{\alpha_1, \alpha_2 \geq 0}{\min} \mathbb{E} \left[ Q^{\boldsymbol{\pi}}(\boldsymbol{s}, \boldsymbol{a}) - \alpha_1(\log(\boldsymbol{\pi}(\boldsymbol{a}|\boldsymbol{s})) + \bar{H}) + \alpha_2(I_u(\boldsymbol{s}, \boldsymbol{a}) - \bar{I}_u(\boldsymbol{s})) \right]$$

Haarnoja et al. (2018) use a simple heuristic $\bar{H} = -\dim(\mathcal{A})$ for the target entropy. However, we cannot specify a general desired information gain since this depends on the learned Bayesian model $p(\boldsymbol{f}^*)$. This makes choosing $\bar{I}_u$ task-dependent. For our experiments, we maintain a target policy $\bar{\boldsymbol{\pi}}$, updated similarly to a target critic in off-policy RL, and define $\bar{I}_u$ as

$$\bar{I}_u(\boldsymbol{s}) := \sum_{j=1}^{d_{\boldsymbol{s}}+1} \mathbb{E}_{\boldsymbol{a} \sim \bar{\boldsymbol{\pi}}(\cdot|\boldsymbol{s})} \left[ \log \left( 1 + \sigma^{-2} \sigma_{n-1,j}^2(\boldsymbol{s}, \boldsymbol{a}) \right) \right] \tag{11}$$

Intuitively, the constraint enforces that the current policy $\boldsymbol{\pi}$ explores w.r.t. the information gain, at least as much as the target policy $\bar{\boldsymbol{\pi}}$. In principle, any other constraint can be used to optimize for $\alpha_2$. We consider our constraint since (*i*) it is easy to evaluate, (*ii*) it can be combined with other intrinsic rewards[1], and (*iii*) it is modular, i.e., it can be added to any RL algorithm. Moreover, as MAXINFORL can be combined with any base off-policy RL algorithm such as SAC (Haarnoja et al., 2018) or DDPG (Lillicrap, 2015), it benefits from the simplicity and scalability of these methods. In addition, it introduces the information gain as a directed exploration bonus and automatically tunes its temperature similar to the policy entropy in Haarnoja et al. (2018). Therefore, it benefits from both the strengths of the naive extrinsic exploration methods and the directedness of intrinsic exploration. We demonstrate this in our experiments, where we combine MAXINFORL with SAC (Haarnoja et al., 2018), REDQ (Chen et al., 2021), DrQ (Yarats et al., 2021), and DrQv2 (Yarats et al., 2022).

**Convergence of MAXINFORL**  In the following, we study our modified Boltzmann exploration strategy and show that as in Haarnoja et al. (2018), the update rules for $Q$ function and the policy converge to an optimal policy $\boldsymbol{\pi}^* \in \Pi$. We make a very general assumption that the entropy of the policy and the model epistemic uncertainty are all bounded for all $(\boldsymbol{s}, \boldsymbol{a}) \in \mathcal{S} \times \mathcal{A}$. The proof of the theorem and the related lemmas are provided in Appendix B.

We define the Bellman operator $\mathcal{T}^{\boldsymbol{\pi}}$

$$\mathcal{T}^{\boldsymbol{\pi}} Q(\boldsymbol{s}, \boldsymbol{a}) = r(\boldsymbol{s}, \boldsymbol{a}) + \gamma \mathbb{E}_{\boldsymbol{s}'|\boldsymbol{s}, \boldsymbol{a}}[V^{\boldsymbol{\pi}}(\boldsymbol{s}')], \tag{12}$$

where

$$V^{\boldsymbol{\pi}}(\boldsymbol{s}) = \mathbb{E}_{\boldsymbol{a} \sim \boldsymbol{\pi}(\cdot|\boldsymbol{s})}[Q(\boldsymbol{s}, \boldsymbol{a}) - \alpha_1 \log(\boldsymbol{\pi}(\boldsymbol{a}|\boldsymbol{s})) + \alpha_2 I_u(\boldsymbol{s}, \boldsymbol{a})] \tag{13}$$

is the soft-value function.

**Theorem 3.1** (MAXINFORL soft Q learning). *Assume that the reward, the entropy for all $\boldsymbol{\pi} \in \Pi$, and the model epistemic uncertainty $\boldsymbol{\sigma}_n$ are all bounded for all $n \geq 0$, $(\boldsymbol{s}, \boldsymbol{a}) \in \mathcal{S} \times \mathcal{A}$. The repeated application of soft policy evaluation (Eq. (12)) and soft policy update (Eq. (9)) to any $\boldsymbol{\pi} \in \Pi$ converges to $\boldsymbol{\pi}^* \in \Pi$ such that $Q^{\boldsymbol{\pi}}(\boldsymbol{s}, \boldsymbol{a}) \leq Q^{\boldsymbol{\pi}^*}(\boldsymbol{s}, \boldsymbol{a})$ for all $\boldsymbol{\pi} \in \Pi$, $(\boldsymbol{s}, \boldsymbol{a}) \in \mathcal{S} \times \mathcal{A}$.*

Theorem 3.1 shows that our reformulated expression for Boltzmann exploration exhibits the same convergence properties from Haarnoja et al. (2018).

## 4 EXPERIMENTS

We evaluate MAXINFORL with Boltzmann exploration from Section 3.2 across several deep RL benchmarks (Brockman, 2016; Tassa et al., 2018; Sferrazza et al., 2024) on state-based and visual control tasks. In all our experiments, we report the mean performance with standard error evaluated over five seeds. For the state-based tasks we combine MAXINFORL with SAC (Haarnoja et al., 2018) and for the visual control tasks with DrQ (Yarats et al., 2021) and DrQv2 (Yarats et al., 2022). In the following, we refer to these algorithms as MAXINFOSAC, MAXINFODRQ, and MAXINFODRQV2, respectively. To further demonstrate the generality of MAXINFORL, in Appendix D, we provide additional experiments, where we combine MAXINFORL with REDQ (Chen et al., 2021), OAC (Ciosek et al., 2019), DrM (Xu et al., 2024), use RND (Burda et al., 2018) as

---

[1]$I$ could represent a different intrinsic reward function, e.g., RND Burda et al. (2018).

an intrinsic reward instead of the information gain, and also evaluate the $\epsilon$–MAXINFORL from Section 3.1 with curiosity (Pathak et al., 2017) and disagreement (Pathak et al., 2019; Sekar et al., 2020; Mendonca et al., 2021) as intrinsic rewards.

**Baselines**: For the state-based tasks, in addition to SAC, we consider four baselines, all of which use SAC as the base RL algorithm:

1. DISAGREEMENT: Employs an explore then exploit strategy, where it maximizes only the intrinsic reward for the first $25\%$ of environment interaction and then switches to the exploitation phase where the extrinsic reward is maximized. We use disagreement in the forward dynamics model for the intrinsic reward.

2. CURIOSITY: The same as DISAGREEMENT but with curiosity as the intrinsic reward.

3. SACINTRINSIC: Based on Burda et al. (2018), where a normalized intrinsic reward is added to the extrinsic reward.

4. SACEIPO: Uses a weighted sum of intrinsic and extrinsic rewards, where the weight is tuned with the extrinsic optimality constraint from Chen et al. (2022).

For the visual control tasks, we use DrQ and DrQv2 as our baselines. More details on our baselines and experiment details are provided in Appendix E.

**Does MAXINFOSAC achieve better performance on state-based control problems?** In Figure 2 we compare MAXINFOSAC with the baselines across several tasks of varying dimensionality[2] from the DeepMind control (DMC, Tassa et al., 2018) and OpenAI gym benchmark suite (Brockman, 2016). Across all tasks we observe that MAXINFOSAC consistently performs the best. While the other baselines perform on par with MAXINFOSAC on some tasks, they fail to solve others. On the contrary, MAXINFOSAC consistently achieves the highest performance in all environments (cf., Fig. 1). To further demonstrate the scalability of MAXINFOSAC, we evaluate it on a practical robotics benchmark, namely HumanoidBench (Sferrazza et al., 2024) that features a simulated Unitree H1 robot on a variety of tasks. We use the *no-hand* version of the benchmark, and evaluate our algorithm on the stand, walk, and run tasks. We compare MAXINFOSAC to SAC in Fig. 3 and again observe that MAXINFOSAC achieves overall higher performance, except for a minor convergence delay on the stand task, which is a trivial exploration problem concerning pure stabilization.

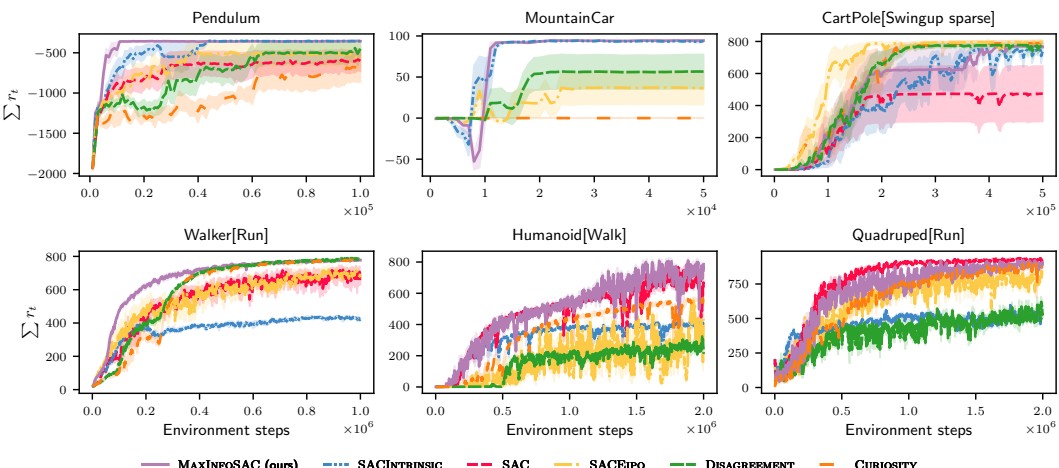

Figure 2: Learning curves of all methods on several environments from the OpenAI gym and DMC suite benchmarks.

**Does MAXINFOSAC solve hard exploration problems?** Naive exploration techniques often struggle with challenging exploration tasks (Burda et al., 2018; Curi et al., 2020; Sukhija et al., 2024b). To test MAXINFOSAC in this context, similar to Curi et al. (2020), we modify the reward in Pendulum, CartPole, and Walker by adding an action cost, $r_{\text{action}}(\boldsymbol{a}) = -K \|\boldsymbol{a}\|_2$, where $K$ controls the penalty for large actions. Curi et al. (2020) empirically show that even for small $K$ values, naive exploration methods fail, converging to the sub-optimal solution of applying no actions.

---

[2]including the humanoid from DMC: $d_s = 67$, $d_a = 21$

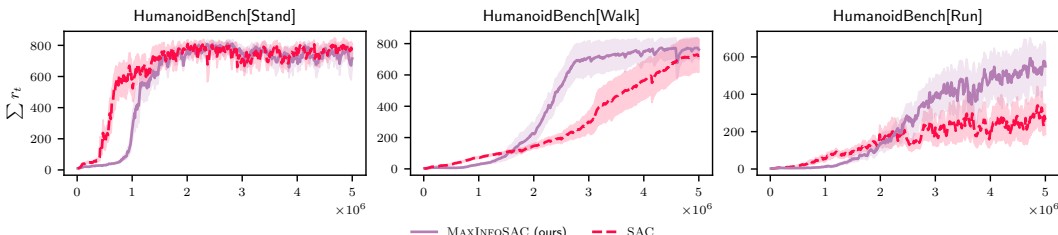

Figure 3: Performance of MaxInfoSAC and SAC on the HumanoidBench benchmark.

In Figure 5 we compare MaxInfoSAC with the baselines. We observe that SAC struggles with action costs, especially in CartPole and Walker. Both SACEipo and SACIntrinsic underperform, likely due to poor handling of extrinsic and intrinsic rewards. Specifically, SACEipo quickly reduces its intrinsic reward weight to zero (cf., Fig. 8 in Appendix D), making it overly greedy. Disagreement and curiosity-based methods perform better since we manually tune their number of intrinsic exploration interactions. However, MaxInfoSAC achieves the best performance by automatically balancing intrinsic and extrinsic rewards. For MaxInfoSAC and SAC, we also depict the phase plot from the exploration on the pendulum environment in Figure 4. MaxInfoSAC covers the state space much faster than SAC, effectively solving the Pendulum swing-up task (Target $= (0, 0)$) within 10K environment steps.

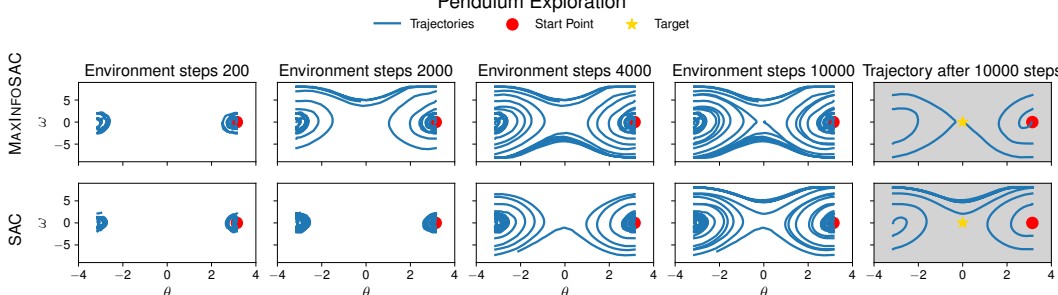

Figure 4: Phase plots during learning of MaxInfoRL and SAC on the Pendulum environment. MaxInfoSAC covers the state space much faster and effectively solves the swing-up task within 10K environment interactions.

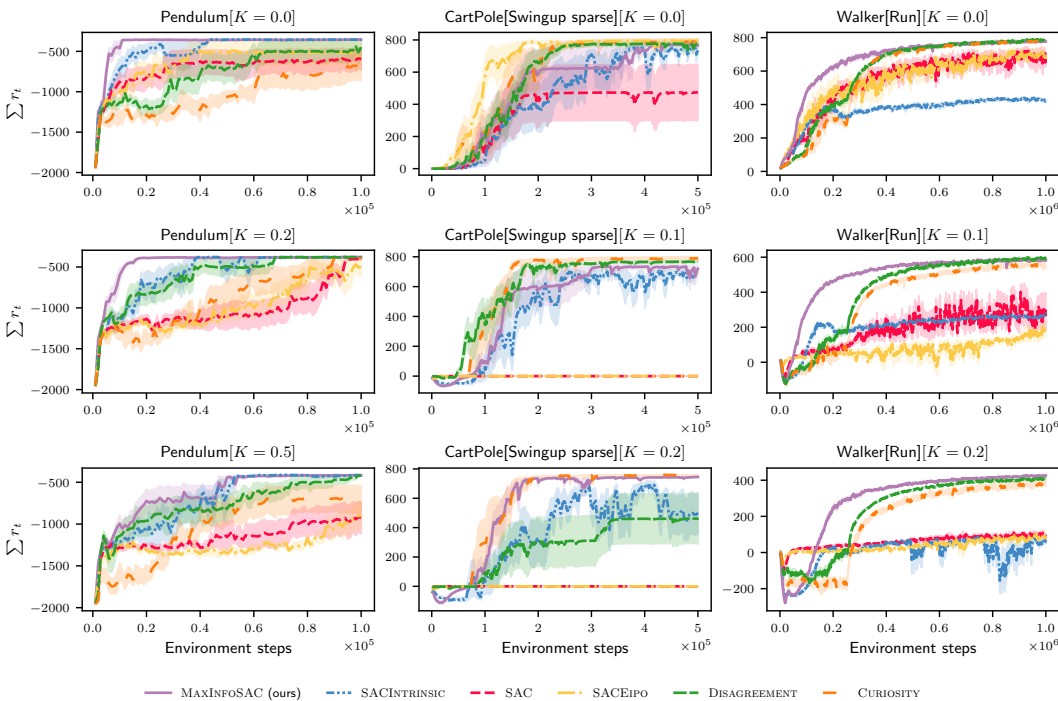

Figure 5: Learning curves for state-based tasks for different values of the action cost parameter $K$.

**Does MAXINFORL scale to visual control tasks?** In this section, we combine MAXINFORL with

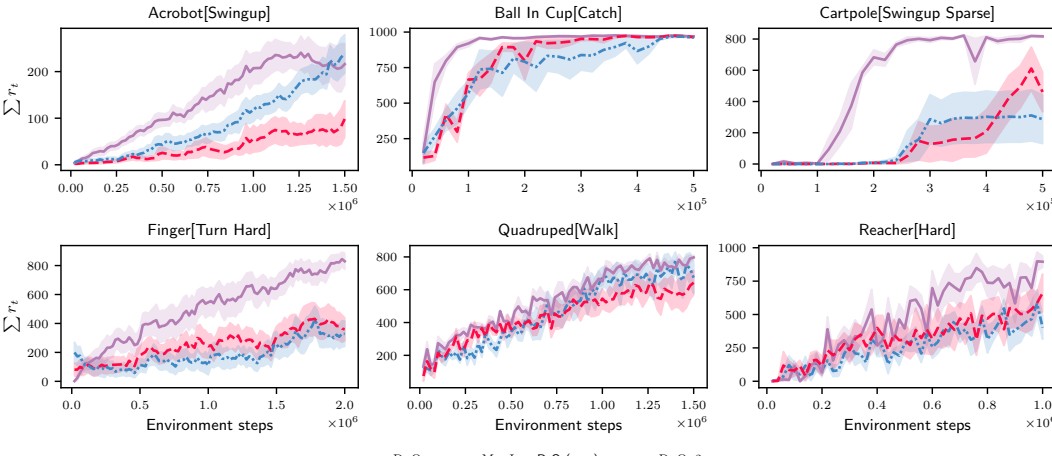

Figure 6: Learning curves from visual control tasks of the DMC suite.

DrQ and evaluate it on visual control problems from the DeepMind control suite (DMC, Tassa et al., 2018). DrQ is a visual control algorithm based on the max entropy framework from SAC, therefore, it can easily be combined with MAXINFORL. We call the resulting algorithm MAXINFODRQ. We compare MAXINFODRQ with DrQ and DrQv2 in Figure 6. From the figure, we conclude that MAXINFODRQ consistently reaches higher rewards and better sample efficiency than the baselines across all tasks. This illustrates the scalability and generality of MAXINFORL.

**Solving challenging visual control tasks with MAXINFORL** For challenging visual control problems, in particular, the humanoid tasks from DMC, Yarats et al. (2022) propose DrQv2, a modified version of DrQ, which uses $n$–step returns and DDPG with noise scheduling instead of SAC for the base algorithm. They claim that switching to DDPG with noise scheduling is particularly useful for solving the humanoid tasks. To this end, we combine MAXINFORL with DrQv2 (MAXINFODRQV2) and evaluate it on the stand, walk, and run humanoid tasks from DMC. This demonstrates the flexibility of MAXINFORL, as it can seamlessly be combined with DrQv2, cf., Appendix E for more details and Appendix D for the performance of MAXINFODRQV2 on other DMC tasks. We compare MAXINFODRQV2 with DrQv2 in Figure 7. MAXINFODRQV2 results in substantial gains in performance and sample efficiency compared to DrQv2. To our knowledge, those shown in Figure 7 are the highest returns reached in these challenging visual control tasks by model-free RL algorithms in the literature. This further illustrates the advantage of directed exploration via information gain/intrinsic rewards.

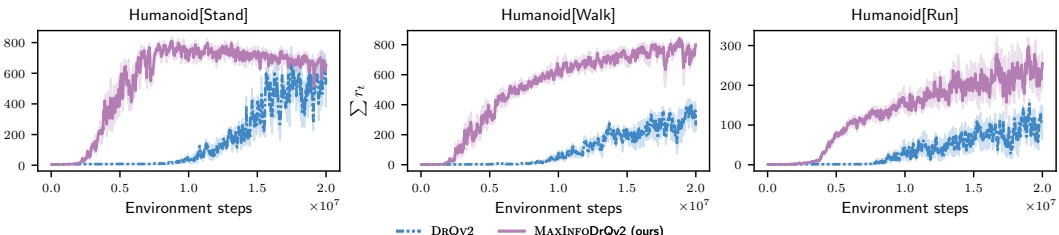

Figure 7: Learning curves from the visual control humanoid tasks of the DMC suite.

# 5 RELATED WORKS

**Naive Exploration** Naive exploration approaches such as $\epsilon$–greedy or Boltzmann are widely applied in RL due to their simplicity (Mnih, 2013; Schulman et al., 2017; Lillicrap, 2015; Haarnoja et al., 2018; Hafner et al., 2023). In particular, the maximum entropy framework (Ziebart et al., 2008) is the basis of many sample-efficient model-free deep RL algorithms (Haarnoja et al., 2018; Chen et al., 2021; Hiraoka et al., 2022; Yarats et al., 2021). However, these methods often perform suboptimally, especially in challenging exploration problems such as those with sparse rewards or local optima (cf., Section 4). Effectively, the agent explores the underlying MDP by taking random sequences of actions. In continuous spaces, this makes sufficiently covering the state and action space exceptionally challenging. Moreover, even in the simplest setting of MAB in continuous

spaces, the most common and theoretically sound exploration strategies are Thompson sampling (TS), upper-confidence bound (UCB) (Srinivas et al., 2012; Chowdhury & Gopalan, 2017) and information-directed sampling (Russo & Van Roy, 2018; Kirschner, 2021). There are several RL algorithms based on these strategies (Brafman & Tennenholtz, 2002; Jaksch et al., 2010; Ouyang et al., 2017; Nikolov et al., 2019; Ciosek et al., 2019; Kakade et al., 2020; Curi et al., 2020; Russo & Proutiere, 2023; Sukhija et al., 2024a). Similarly, there are methods from Bayesian RL (Osband et al., 2018; Fellows et al., 2021; Buening et al., 2023) that perform principled exploration. However, the naive exploration approaches remain ubiquitous in deep RL due to their simplicity. Instead, intrinsic rewards are often used to facilitate more directed exploration. However, how to combine extrinsic and intrinsic exploration is much less understood both theoretically and practically.

**Intrinsic exploration** Several works use intrinsic rewards as a surrogate objective to facilitate exploration in challenging tasks (cf., Aubret et al., 2023, for a comprehensive survey). Common choices of intrinsic rewards are model prediction error or "Curiosity" (Schmidhuber, 1991; Pathak et al., 2017; Burda et al., 2018), novelty of transitions/state-visitation counts (Stadie et al., 2015; Bellemare et al., 2016), diversity of skills/goals (Eysenbach et al., 2018; Sharma et al., 2019; Nair et al., 2018; Pong et al., 2019), empowerment (Klyubin et al., 2005; Salge et al., 2014), state-entropy (Mutti et al., 2021; Seo et al., 2021; Kim et al., 2024), and information gain over the forward dynamics (Sekar et al., 2020; Mendonca et al., 2021; Sukhija et al., 2024b). In this work, we focus on the information gain since it is the basis of many theoretically sound and empirically strong active learning methods (Krause et al., 2008; Balcan et al., 2010; Hanneke et al., 2014; Mania et al., 2020; Sekar et al., 2020; Sukhija et al., 2024b; Hübotter et al., 2024). Furthermore, we also motivate the choice of information gain/model epistemic uncertainty theoretically (cf., Theorem A.1 and Appendix C). Nonetheless, MAXINFORL can also be used with other intrinsic exploration objectives. In this work, we study how to combine the intrinsic rewards with the extrinsic ones for exploration in RL. Moreover, approaches such as Pathak et al. (2017; 2019); Sukhija et al. (2024b); Sekar et al. (2020) execute an explore then exploit strategy, where initially the extrinsic reward is ignored and a policy is trained to maximize the intrinsic objective. After this initial exploration phase, the agent is trained to maximize the extrinsic reward. This strategy is common in active learning methods and does not result in sublinear cumulative regret even for the simple MAB setting. This is because the agent executes a finite number of exploration steps instead of continuously trading off exploration and exploitation. To this end, Burda et al. (2018); Chen et al. (2022) combine the extrinsic and intrinsic exploration rewards by taking a weighted sum $r_{\text{tot}} = \lambda r_{\text{extrinsic}} + r_{\text{intrinsic}}$. While Burda et al. (2018) use fixed weight $\lambda$, Chen et al. (2022) propose the extrinsic optimality constraint to auto-tune the intrinsic reward weight. The extrinsic optimality constraint enforces that the policy maximizing the sum of intrinsic and extrinsic rewards performs at least as well as the policy which only maximizes the extrinsic ones. Keeping a fixed weight $\lambda$ across all tasks and environment steps might be suboptimal and the constraint from Chen et al. (2022) may quickly lead to the trivial greedy solution $\lambda^{-1} = 0$ (cf., Fig. 8 in Appendix D). We compare MAXINFORL with these two methods, as well as with explore then exploit strategies. We show that MAXINFORL outperforms them across several deep RL benchmarks (cf., Section 4).

## 6 CONCLUSION

In this work, we introduced MAXINFORL, a class of model-free off-policy RL algorithms that train an agent to trade-off reward maximization with an exploration objective that targets high entropy in the state, action, and reward spaces. The proposed approach is theoretically sound, simple to implement, and can be combined with most common RL algorithms. MAXINFORL consistently outperforms its baselines in a multitude of benchmarks and achieves state-of-the-art performance on challenging visual control tasks.

A limitation of MAXINFORL is that it requires training an ensemble of forward dynamics models to compute the information gain, which increases computation overhead (c.f. Appendix E Table 1). In addition, while effective, the constraint in Equation (10) also requires maintaining a target policy.

The generality of the MAXINFORL exploration bonus and the trade-off we propose with reward maximization are not limited to the off-policy model-free setting, which we focus on here due to their sample and computational efficiency. Future work will investigate its applicability to other classes of RL algorithms, such as model-based RL, where the forward dynamics model is generally part of the training framework. In fact, in this work we only use the forward dynamics model for the intrinsic rewards. Another interesting direction is to include samples from the learned model in the policy training, which may yield additional gains in sample efficiency. Lastly, extending our theoretical guarantees from the bandit settings to the MDP case is also an interesting direction for future work.

ACKNOWLEDGMENTS

The authors would like to thank Kevin Zakka for the helpful discussions. This project was supported in part by the ONR Science of Autonomy Program N000142212121, the Swiss National Science Foundation under NCCR Automation, grant agreement 51NF40 180545, the Microsoft Swiss Joint Research Center, the SNSF Postdoc Mobility Fellowship 211086, and an ONR DURIP grant. Pieter Abbeel holds concurrent appointments as a Professor at UC Berkeley and as an Amazon Scholar. This paper describes work performed at UC Berkeley and is not associated with Amazon.

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

# A    ANALYZING $\epsilon$–GREEDY FOR MULTI-ARMED BANDITS

In this section, we enunciate a theorem that shows that our modified $\epsilon$–greedy approach from Section 3.1 has sublinear regret in the simplified setting of multi-arm bandits.

## A.1    SUBLINEAR REGRET FOR $\epsilon$–MAXINFORL

This exploration–exploitation trade-off is also fundamental in multi-armed bandits (MAB) (Lattimore & Szepesvári, 2020), where the task is to optimize an unknown objective function $J : \Theta \to [-\frac{1}{2}J_{\max}, \frac{1}{2}J_{\max}]$, with $\Theta \subset \mathbb{R}^d$ being a compact set over which we seek the optimal arm $\boldsymbol{\theta}^* = \arg\max_{\boldsymbol{\theta}\in\Theta} J(\boldsymbol{\theta})$. In each round $t$, we choose a point $\boldsymbol{\theta}_t \in \Theta$ and obtain a noisy measurement $y_t$ of the function value, that is, $y_t = J(\boldsymbol{\theta}_t) + w_t$. Our goal is to maximize the sum of rewards $\sum_{t=1}^{T} J(\boldsymbol{\theta}_t)$ over $T$ learning iterations/episodes, thus to perform essentially as well as $\boldsymbol{\theta}^*$ (as quickly as possible). For example, $\boldsymbol{\theta}$ can be parameters of the policy distribution $\boldsymbol{\pi_\theta}$ and our objective maximizing the discounted rewards as defined in Eq. (1), i.e., $\max_{\boldsymbol{\theta}\in\Theta} J(\boldsymbol{\pi_\theta})$. This formulation is the basis of several sample-efficient RL algorithms (Calandra et al., 2016; Marco et al., 2016; Antonova et al., 2017; Berkenkamp et al., 2021; Sukhija et al., 2023).

The natural performance metric in this context is the cumulative regret defined as $R_T = \sum_{t=1}^{T} J(\boldsymbol{\theta}^*) - J(\boldsymbol{\theta}_t)$. A desirable asymptotic property of any learning algorithm is that it is no-regret: $\lim_{T\to\infty} R_T/T = 0$. In the following theorem, we show that under standard continuity assumptions on $J$, i.e., we can model it through Gaussian process regression (Rasmussen & Williams, 2005) with a given kernel $k(\cdot, \cdot) : \Theta \times \Theta \to \mathbb{R}_+$, our exploration strategy from Eq. (7) with the model epistemic uncertainty as the intrinsic objective has sublinear regret.

**Theorem A.1.** *Assume that the objective function $J$ lies in the RKHS $\mathcal{H}_k$ corresponding to the kernel $k(\boldsymbol{\theta}, \boldsymbol{\theta})$, $\|J\|_k \leq B$, and that the noise $w_t$ is zero-mean $\sigma$-sub Gaussian. Let $\epsilon_t = t^{2\alpha-1}$ for $\alpha \in (1/4, 1/2)$, $\delta \in (0, 1]$ and consider the following $\epsilon$–greedy exploration strategy*

$$\boldsymbol{\theta}_t = \begin{cases} \arg\max_{\boldsymbol{\theta}\in\Theta} \sigma_t(\boldsymbol{\theta}) & \text{with probability} \epsilon_t \\ \arg\max_{\boldsymbol{\theta}\in\Theta} \mu_t(\boldsymbol{\theta}) & \text{else,} \end{cases}$$

*where $\mu_t$ is our mean estimate of $J$ and $\sigma_t$ the epistemic uncertainty (c.f., (Rasmussen & Williams, 2005) for the exact formula). Then we have with probability at least $1 - \delta$*

$$R_T \leq \mathcal{O}\left(J_{\max}T^{2\alpha} + 2\Gamma_T(k)T^{1-\alpha}\right),$$

*where $\Gamma_T(k)$ is the maximum information gain (Srinivas et al., 2012).*

The maximum information gain $\Gamma_T(k)$ measures the complexity of learning the function $J$ w.r.t. the number of data points $T$ and depends on the choice of the kernel $k$, for common kernels such as the RBF and linear kernel, it grows polylogarithmically with $T$ (Srinivas et al., 2012; Vakili et al., 2021), resulting in a sublinear regret bound. Intuitively, by sampling informative actions sufficiently often, we will improve our estimate of $J$ and thus gradually suffer less regret.

## A.2    PROOF OF THEOREM A.1

From hereon, let $Z_t \sim \text{Bernoulli}(\epsilon_t)$ be a Bernoulli random variable. The acquisition function for $\epsilon$-greedy GP bandit from Theorem A.1 is defined as:

$$\boldsymbol{\theta}_t = \begin{cases} \arg\max_{\boldsymbol{\theta}\in\Theta} \sigma_t(\boldsymbol{\theta}) & \text{if } Z_t = 1 \\ \arg\max_{\boldsymbol{\theta}\in\Theta} \mu_t(\boldsymbol{\theta}) & \text{else} \end{cases} \tag{14}$$

Let $Q_t = \sum_{s=1}^{t} Z_s - \epsilon_s$. In the following, we analyze the sequence $Q_{1:t}$. Note that $Q_t \in [0, t]$ for all $t$ and

$$\begin{aligned} \mathbb{E}[Q_t | Q_{1:t-1}] &= \mathbb{E}[Z_t - \epsilon_t] + \mathbb{E}[Q_{t-1} | Q_{1:t-1}] \\ &= Q_{t-1}. \end{aligned}$$

Therefore, $Q_{1:t}$ is a martingale sequence, or equivalently, $\{Z_s - \epsilon_s\}_{s\in 1:t}$ is a martingale difference sequence.

In the following, we use the time-uniform Azuma Hoeffding inequality from Kassraie et al. (2024) to give a bound on $\sum_{s=1}^{T} Z_s$.

**Lemma A.2.** *Let $\epsilon_1 = 1$, then we have with probability at least $1 - \delta$, for all $t \geq 1$*

$$\sum_{s=1}^{t} Z_s \geq \max \left\{ \sum_{s=1}^{t} \epsilon_s - \frac{5}{2}\sqrt{t(\log\log(et) + \log(2\delta))}, 1 \right\}$$

*Proof.* Note that $|Q_t - Q_{t-1}| = |Z_t - \epsilon_t| \in [0, 1]$ for all $t$. Therefore, we can use the time-uniform Azzuma-Hoeffding inequality (Kassraie et al., 2024, Lemma 26.) to get

$$\Pr\left( \exists t : \sum_{s=1}^{t} \epsilon_s - Z_s \geq \frac{5}{2}\sqrt{t(\log\log(et) + \log(2\delta))} \right) \leq \delta$$

Therefore, we have with probability at least $1 - \delta$

$$\sum_{s=1}^{t} Z_s \geq \sum_{s=1}^{t} \epsilon_s - \frac{5}{2}\sqrt{t(\log\log(et) + \log(2\delta))}$$

for all $t \geq 1$. Furthermore, since $\epsilon_1 = 1$, we have $Z_1 = 1$. Therefore,

$$\sum_{s=1}^{t} Z_s \geq \max \left\{ \sum_{s=1}^{t} \epsilon_s - \frac{5}{2}\sqrt{t(\log\log(et) + \log(2\delta))}, 1 \right\}$$

$\square$

Let $E_t$ denote the number of exploration steps after $t$ iterations of the algorithm, that is, $E_t = \sum_{s=1}^{t} Z_s$. In the next lemma, we derive a bound on $E_t$.

**Lemma A.3.** *Let $\epsilon_t = t^{2\alpha-1}$ with $\alpha \in (1/4, 1/2)$, then for any $\delta \in (0, 1]$ exists a $C > 0, t_0 > 0$ such that $E_t \geq C^2 t^{2\alpha}$ for all $t \geq t_0$ with probability at least $1 - \delta$.*

*Proof.* Note that $\epsilon_t$ is monotonously decreasing with $t$. Therefore, $\int_s^{s+1} x^{2\alpha-1} dx \leq \epsilon_s$ and hence

$$\sum_{s=1}^{t} \epsilon_s \geq \int_1^t x^{2\alpha-1} dx = \frac{t^{2\alpha} - 1}{2\alpha}$$

Therefore, we get

$$\sum_{s=1}^{t} \epsilon_s - \frac{5}{2}\sqrt{t(\log\log(et) + \log(2\delta))} \geq \frac{t^{2\alpha}}{2\alpha} - \left( \frac{5}{2}\sqrt{t(\log\log(et) + \log(2\delta))} + \frac{1}{2\alpha} \right)$$

Therefore,

$$E_t = \sum_{s=1}^{t} Z_s$$

$$\geq \max \left\{ \sum_{s=1}^{t} \epsilon_s - \frac{5}{2}\sqrt{t(\log\log(et) + \log(2\delta))}, 1 \right\}$$

$$\geq \max \left\{ \frac{t^{2\alpha}}{2\alpha} - \left( \frac{5}{2}\sqrt{t(\log\log(et) + \log(2\delta))} + \frac{1}{2\alpha} \right), 1 \right\}$$

Note that since $\alpha > \frac{1}{4}$, for any $\delta \in (0, 1]$

$$\frac{5}{2}\sqrt{t(\log\log(et) + \log(2\delta))} + \frac{1}{2\alpha} \in o(t^{2\alpha}).$$

Moreover $\max \left\{ \frac{t^{2\alpha}}{2\alpha} - \left( \frac{5}{2}\sqrt{t(\log\log(et) + \log(2\delta))} + \frac{1}{2\alpha} \right), 1 \right\} \in \Theta(t^{2\alpha})$, therefore for any $\delta \in (0, 1]$ exists a $C > 0, t_0 > 0$ such that $E_t \geq C^2 t^{2\alpha}$ for all $t \geq t_0$ with probability at least $1 - \delta$.

$\square$

Next, we use the results from Chowdhury & Gopalan (2017) on the well-calibration of $J$ w.r.t. our mean and epistemic uncertainty estimates.

**Lemma A.4** (Theorem 2, Chowdhury & Gopalan (2017)). *Let the assumptions from Theorem A.1 hold. Then, with probability at least $1 - \delta$, the following holds for all $\boldsymbol{\theta} \in \Theta$ and $t \geq 1$:*

$$|\mu_{t-1}(\boldsymbol{\theta}) - J(\boldsymbol{\theta})| \leq \beta_{t-1}\sigma_{t-1}(\boldsymbol{\theta}),$$

*with $\beta_{t-1} = B + \sigma\sqrt{2(\Gamma_{t-1} + 1 + \ln(1/\delta))}$, where $\Gamma_{t-1}$ is the maximum information gain (Srinivas et al., 2012) after $t - 1$ rounds.*

*Proof of Theorem A.1.* The $\epsilon$–greedy algorithm alternates between uncertainty sampling and maximizing the mean. In the worst case, it may suffer maximal regret $J_{\max}$ during the uncertainty sampling stage. Furthermore, assume at time $t$, we execute the exploitation stage, then we have

$$
\begin{aligned}
J(\boldsymbol{\theta}^*) - J(\boldsymbol{\theta}_t) &= J(\boldsymbol{\theta}^*) - \mu_t(\boldsymbol{\theta}^*) + \mu_t(\boldsymbol{\theta}^*) - J(\boldsymbol{\theta}_t) \\
&\leq J(\boldsymbol{\theta}^*) - \mu_t(\boldsymbol{\theta}^*) + \mu_t(\boldsymbol{\theta}_t) - J(\boldsymbol{\theta}_t) && (\mu_t(\boldsymbol{\theta}_t) \geq \mu_t(\boldsymbol{\theta}^*)) \\
&\leq \beta_t(\sigma_t(\boldsymbol{\theta}^*) + \sigma_t(\boldsymbol{\theta}_t)) && \text{(Lemma A.4)}
\end{aligned}
$$

Therefore, we can break down the cumulative regret $R_T$ into these two stages.

$$
\begin{aligned}
R_T &= \mathbb{E}_{Z_{1:T},w_{1:T}}\left[\sum_{t=1}^{T} J(\boldsymbol{\theta}^*) - J(\boldsymbol{\theta}_t)\right] \\
&= \sum_{t=1}^{T}[\mathbb{E}_{Z_{1:t-1},w_{1:t-1}}\mathbb{E}_{w_t,Z_t}[J(\boldsymbol{\theta}^*) - J(\boldsymbol{\theta}_t)]] \\
&= \sum_{t=1}^{T}\mathbb{E}_{Z_{1:t-1},w_{1:t-1}}[\mathbb{E}_{w_t}[J(\boldsymbol{\theta}^*) - J(\boldsymbol{\theta}_t)|Z_t=0](1-\epsilon_t) + \mathbb{E}_{w_t}[J(\boldsymbol{\theta}^*) - J(\boldsymbol{\theta}_t)|Z_t=1]\epsilon_t \\
&\leq J_{\max}\sum_{t=1}^{T}\epsilon_t + \sum_{t=1}^{T}\beta_t\mathbb{E}_{Z_{1:t-1},w_{1:t-1}}[\sigma_t(\boldsymbol{\theta}^*) + \sigma_t(\boldsymbol{\theta}_t)]
\end{aligned}
$$

Assume $E_t$ is given, then we have for all $t$

$$
\begin{aligned}
\max_{\boldsymbol{\theta}\in\Theta}\sigma_t^2(\boldsymbol{\theta})E_t &= \max_{\boldsymbol{\theta}\in\Theta}\sigma_t^2(\boldsymbol{\theta})\sum_{s=1}^{t}Z_s \\
&\leq \sum_{s=1}^{t}Z_s\max_{\boldsymbol{\theta}\in\Theta}\sigma_s^2(\boldsymbol{\theta}) \\
&\leq \sum_{s=1}^{t}Z_s\sigma_s^2(\boldsymbol{\theta}_s) \\
&\leq \sum_{s=1}^{t}\sigma_s^2(\boldsymbol{\theta}_s) \\
&\leq C\Gamma_t
\end{aligned}
$$

Therefore, $\sigma_t^2(\boldsymbol{\theta}) \leq \frac{C\Gamma_t}{E_t}$ for all $\boldsymbol{\theta} \in \mathcal{X}$. Since, $E_t \geq C^2 t^{2\alpha}$ for $\alpha \in \left(\frac{1}{4}, \frac{1}{2}\right)$, we have with probability at least $1 - \delta$ for all $t \geq t_0$, $\boldsymbol{\theta} \in \Theta$, there exists a constant $K$ such that

$$\sigma_t(\boldsymbol{\theta}) \leq K\frac{\sqrt{\Gamma_t}}{t^\alpha}$$

Going back to the regret, we get

$$
\begin{aligned}
R_t &\leq J_{\max}\sum_{t=1}^{T}\epsilon_t + \sum_{t=1}^{T}2\beta_t K\frac{\sqrt{\Gamma_t}}{t^\alpha} \\
&\leq \mathcal{O}\left(J_{\max}T^{2\alpha} + 2K\beta_T\sqrt{\Gamma_T}T^{1-\alpha}\right)
\end{aligned}
$$

Here we used the fact that, $\alpha \in \left(\frac{1}{4}, \frac{1}{2}\right)$, therefore,

$$\sum_{t=1}^{T} 2\beta_t K \frac{\sqrt{\Gamma_t}}{t^\alpha} \leq 2K\beta_T \sqrt{\Gamma_T} \sum_{t=1}^{T} \frac{1}{t^\alpha} \leq \Theta(2K\beta_T \sqrt{\Gamma_T} T^{1-\alpha}).$$

Similarly since $\epsilon_t = t^{2\alpha-1}$, we have $\sum_{t=1}^{T} \epsilon_t = \Theta(T^{2\alpha})$ The regret is sublinear for ubiquitous choices of kernels such as the RBF. Indicating, consistency of the $\epsilon$-greedy exploration strategy with the model-epistemic uncertainty as the exploration signal.

$\square$

## B  PROOF OF THEOREM 3.1

**Lemma B.1.** *Consider the Bellman operator $\mathcal{T}^{\boldsymbol{\pi}}$ from Eq. (12), and a mapping $Q^0 : \mathcal{S} \times \mathcal{A} \to \mathbb{R}$ and define $Q^{k+1} = \mathcal{T}^{\boldsymbol{\pi}} Q^k$. Furthermore, let the assumptions from Theorem 3.1 hold. Then, the sequence $Q^k$ will converge to the soft-Q function of $\boldsymbol{\pi}$ as $k \to \infty$.*

*Proof.* Consider the following augmented reward

$$r_{\boldsymbol{\pi}}(\boldsymbol{s}, \boldsymbol{a}) = r(\boldsymbol{s}, \boldsymbol{a}) + \mathbb{E}_{\boldsymbol{s}'|\boldsymbol{s},\boldsymbol{a}} \left[ \mathbb{E}_{\boldsymbol{a}' \sim \boldsymbol{\pi}(\cdot|\boldsymbol{s}')}[-\alpha_1 \log(\boldsymbol{\pi}(\boldsymbol{a}'|\boldsymbol{s}')) + \alpha_2 I(\tilde{\boldsymbol{s}}''; \boldsymbol{f}^*|\boldsymbol{s}', \boldsymbol{a}')] \right],$$

where $\tilde{\boldsymbol{s}}''$ is the next state and reward given $(\boldsymbol{s}', \boldsymbol{a}')$. Then from Eq. (12), we have

$$\mathcal{T}^{\boldsymbol{\pi}} Q(\boldsymbol{s}, \boldsymbol{a}) = r_{\boldsymbol{\pi}}(\boldsymbol{s}, \boldsymbol{a}) + \gamma \mathbb{E}_{\boldsymbol{s}',\boldsymbol{a}'|\boldsymbol{s},\boldsymbol{a}}[Q(\boldsymbol{s}', \boldsymbol{a}')].$$

Furthermore, $r_{\boldsymbol{\pi}}$ is bounded since $r$ is bounded and the policy entropy and the information gain are bounded for all $\boldsymbol{\pi} \in \Pi$ by assumption. Therefore, we can apply standard convergence results on policy evaluation to show convergence (Sutton, 2018). $\square$

Lemma B.1 shows that the Q updates satisfy the contraction property and thus converge to the soft-Q function of $\boldsymbol{\pi}$. Next, we show that the policy update from Eq. (8) leads to monotonous improvement of the policy.

**Lemma B.2.** *Let the assumptions from Theorem 3.1 hold. Consider any $\boldsymbol{\pi}^{old} \in \Pi$, and let $\boldsymbol{\pi}^{new}$ denote the solution to Eq. (8). Then we have for all $(\boldsymbol{s}, \boldsymbol{a}) \in \mathcal{S} \times \mathcal{A}$ that $Q^{\boldsymbol{\pi}^{old}}(\boldsymbol{s}, \boldsymbol{a}) \leq Q^{\boldsymbol{\pi}^{new}}(\boldsymbol{s}, \boldsymbol{a})$.*

*Proof.* Consider any $\boldsymbol{s} \in \mathcal{S}$

$$\mathbb{E}_{\boldsymbol{a} \sim \boldsymbol{\pi}^{new}(\cdot|\boldsymbol{s})} \left[ Q^{\boldsymbol{\pi}^{old}}(\boldsymbol{s}, \boldsymbol{a}) - \alpha_1 \log \boldsymbol{\pi}^{new}(\boldsymbol{a}|\boldsymbol{s}) + \alpha_2 I(\boldsymbol{s}'; \boldsymbol{f}^*|\boldsymbol{s}, \boldsymbol{a}) \right]$$
$$\geq \mathbb{E}_{\boldsymbol{a} \sim \boldsymbol{\pi}^{old}(\cdot|\boldsymbol{s})} \left[ Q^{\boldsymbol{\pi}^{old}}(\boldsymbol{s}, \boldsymbol{a}) - \alpha_1 \log \boldsymbol{\pi}^{old}(\boldsymbol{a}|\boldsymbol{s}) + \alpha_2 I(\boldsymbol{s}'; \boldsymbol{f}^*|\boldsymbol{s}, \boldsymbol{a}) \right]$$
$$= V^{\boldsymbol{\pi}^{old}}(\boldsymbol{s}) \tag{15}$$

Next consider any pair $(\boldsymbol{s}, \boldsymbol{a}) \in \mathcal{S} \times \mathcal{A}$

$$Q^{\boldsymbol{\pi}^{old}}(\boldsymbol{s}, \boldsymbol{a}) = r(\boldsymbol{s}, \boldsymbol{a}) + \gamma \mathbb{E}_{\boldsymbol{s}'|\boldsymbol{s},\boldsymbol{a}}[V^{\boldsymbol{\pi}^{old}}(\boldsymbol{s}')]$$
$$\leq r(\boldsymbol{s}, \boldsymbol{a}) + \gamma \mathbb{E}_{\boldsymbol{s}'|\boldsymbol{s},\boldsymbol{a}}[\mathbb{E}_{\boldsymbol{a}' \sim \boldsymbol{\pi}^{new}(\cdot|\boldsymbol{s}')}[Q^{\boldsymbol{\pi}^{old}}(\boldsymbol{s}', \boldsymbol{a}') - \alpha_1 \log \boldsymbol{\pi}^{new}(\boldsymbol{a}'|\boldsymbol{s}') + \alpha_2 I(\boldsymbol{s}''; \boldsymbol{f}^*|\boldsymbol{s}', \boldsymbol{a}')]]$$
$$\vdots$$
$$\leq Q^{\boldsymbol{\pi}^{new}}(\boldsymbol{s}, \boldsymbol{a})$$

where we have repeatedly expanded $Q^{\boldsymbol{\pi}^{old}}$ on the RHS by applying the soft Bellman equation and the bound in Eq. (15). $\square$

Finally, combining Lemma B.1 and Lemma B.2, we can prove Theorem 3.1.

*Proof of Theorem 3.1.* Note that since the reward, information gain, and entropy are bounded, $Q^{\boldsymbol{\pi}}(\boldsymbol{s}, \boldsymbol{a})$ is also bounded for all $\boldsymbol{\pi} \in \Pi$, $(\boldsymbol{s}, \boldsymbol{a}) \in \mathcal{S} \times \mathcal{A}$. Furthermore, in Lemma B.2 we show that the policy is monotonously improving. Therefore, the sequence of policy updates converges to a policy $\boldsymbol{\pi}^* \in \Pi$ such that $J^{\boldsymbol{\pi}^*}(\boldsymbol{\pi}^*) \geq J^{\boldsymbol{\pi}^*}(\boldsymbol{\pi})$ for all $\boldsymbol{\pi} \in \Pi$. Applying the same argument as Haarnoja et al. (2018, Theorem 1), we get $Q^{\boldsymbol{\pi}^*}(\boldsymbol{s}, \boldsymbol{a}) \geq Q^{\boldsymbol{\pi}}(\boldsymbol{s}, \boldsymbol{a})$ for all $\boldsymbol{\pi} \in \Pi$, $(\boldsymbol{s}, \boldsymbol{a}) \in \mathcal{S} \times \mathcal{A}$. $\square$

## C   MAXINFORL FROM THE PERSPECTIVE OF KL-MINIMIZATION

We take inspiration from Hafner et al. (2020) and study the Boltzmann exploration formulation from Section 3.2 from the action perception and divergence minimization (APD) perspective. We denote with $\boldsymbol{\tau} = \{\boldsymbol{s}_t, \boldsymbol{a}_t, r_t\}_{t\geq 0}$ the trajectory in state, reward, and action space. The goal of the RL agent is to maximize the cumulative reward w.r.t. the true system $\boldsymbol{f}^*$. Therefore, a common and natural choice for a target/desired distribution for the trajectory $\boldsymbol{\tau}$ is (Levine, 2018; Hafner et al., 2020)

$$
p^*(\boldsymbol{\tau}) = \left[\prod_{t\geq 0} p(\boldsymbol{s}_{t+1}, r_t|\boldsymbol{s}_t, \boldsymbol{a}_t, \boldsymbol{f}^*)\right] \exp\left(\frac{1}{\alpha}\sum_{t\geq 0} r_t\right).
$$

However, $\boldsymbol{f}^*$ is unknown in our case and we only have an estimate of the underlying distribution $p(\boldsymbol{f})$ from which it is sampled. Given a state-pair $(\boldsymbol{s}_t, \boldsymbol{a}_t)$, we get the following distribution for $(\boldsymbol{s}_{t+1}, r_t)$.

$$
p(\boldsymbol{s}_{t+1}, r_t|\boldsymbol{s}_t, \boldsymbol{a}_t) = \int p(\boldsymbol{s}_{t+1}, r_t|\boldsymbol{s}_t, \boldsymbol{a}_t, \boldsymbol{f})dp(\boldsymbol{f}) = E_{\boldsymbol{f}}\left[p(\boldsymbol{s}_{t+1}, r_t|\boldsymbol{s}_t, \boldsymbol{a}_t, \boldsymbol{f})\right]. \tag{16}
$$

For a given policy $\boldsymbol{\pi}$, we can write the distribution of its trajectory via $\boldsymbol{\tau}^{\boldsymbol{\pi}}$ as

$$
\begin{aligned}
\hat{p}^{\boldsymbol{\pi}}(\boldsymbol{\tau}) &= \left[\prod_{t\geq 0} p(\boldsymbol{s}_{t+1}, r_t|\boldsymbol{s}_t, \boldsymbol{a}_t)\boldsymbol{\pi}(\boldsymbol{a}_t|\boldsymbol{s}_t)\right] \\
&= p(\boldsymbol{s}_1)\left[\prod_{t\geq 0} E_{\boldsymbol{f}}\left[p(\boldsymbol{s}_{t+1}, r_t|\boldsymbol{s}_t, \boldsymbol{a}_t, \boldsymbol{f})\right]\boldsymbol{\pi}(\boldsymbol{a}_t|\boldsymbol{s}_t)\right].
\end{aligned} \tag{17}
$$

A natural objective for deciding which policy to pick is to select $\boldsymbol{\pi}$ such that its resulting trajectory $\boldsymbol{\tau}^{\boldsymbol{\pi}}$ is close (in-distribution) to $\boldsymbol{\tau}^*$. That is,

$$
\boldsymbol{\pi}^* = \arg\min_{\boldsymbol{\pi}} D_{\mathrm{KL}}(\hat{p}^{\boldsymbol{\pi}}(\boldsymbol{\tau})|p^*(\boldsymbol{\tau})). \tag{18}
$$

In the following, we break down the term from Eq. (18) and show that it results in the MAXINFORL objective.

$$
\begin{aligned}
D_{\mathrm{KL}}(\hat{p}^{\boldsymbol{\pi}}(\boldsymbol{\tau})|p^*(\boldsymbol{\tau})) &= \mathbb{E}_{\boldsymbol{\tau}\sim\hat{p}^{\boldsymbol{\pi}}(\boldsymbol{\tau})}\left[-\log\left(\frac{p^*(\boldsymbol{\tau})}{\hat{p}^{\boldsymbol{\pi}}(\boldsymbol{\tau})}\right)\right] \\
&= -\mathbb{E}_{\boldsymbol{\tau}\sim\hat{p}^{\boldsymbol{\pi}}(\boldsymbol{\tau})}\left[\sum_{t\geq 0}\log(p(\boldsymbol{s}_{t+1}, r_t|\boldsymbol{s}_t, \boldsymbol{a}_t, \boldsymbol{f}^*)) + \frac{1}{\alpha}\sum_{t\geq 0} r_t\right] \\
&\quad + \mathbb{E}_{\boldsymbol{\tau}\sim\hat{p}^{\boldsymbol{\pi}}(\boldsymbol{\tau})}\left[\sum_{t\geq 0}\log\left(E_{\boldsymbol{f}}\left[p(\boldsymbol{s}_{t+1}, r_t|\boldsymbol{s}_t, \boldsymbol{a}_t, \boldsymbol{f})\right]\right) + \log(\boldsymbol{\pi}(\boldsymbol{a}_t|\boldsymbol{s}_t))\right] \\
&= \mathbb{E}_{\boldsymbol{\tau}\sim\hat{p}^{\boldsymbol{\pi}}(\boldsymbol{\tau})}\left[\sum_{t\geq 0}\log(\boldsymbol{\pi}(\boldsymbol{a}_t|\boldsymbol{s}_t)) - \frac{1}{\alpha}r_t\right] \\
&\quad + \mathbb{E}_{\boldsymbol{\tau}\sim\hat{p}^{\boldsymbol{\pi}}(\boldsymbol{\tau})}\left[\sum_{t\geq 0}\log\left(E_{\boldsymbol{f}}\left[p(\boldsymbol{s}_{t+1}, r_t|\boldsymbol{s}_t, \boldsymbol{a}_t, \boldsymbol{f})\right]\right) - \log(p(\boldsymbol{s}_{t+1}|\boldsymbol{s}_t, \boldsymbol{a}_t, \boldsymbol{f}^*))\right]
\end{aligned}
$$

$$\tag{1}$$

$$\tag{2}$$

The term (1) is commonly minimized in max entropy RL methods such as Haarnoja et al. (2018). The term (2) has a very natural interpretation, which we discuss below

$$\mathbb{E}_{\boldsymbol{\tau} \sim \hat{p}^{\boldsymbol{\pi}}(\boldsymbol{\tau})} \left[ \sum_{t \geq 0} \log \left( E_{\boldsymbol{f}} \left[ p(\boldsymbol{s}_{t+1}, r_t | \boldsymbol{s}_t, \boldsymbol{a}_t, \boldsymbol{f}) \right] \right) - \log(p(\boldsymbol{s}_{t+1}, r_t | \boldsymbol{s}_t, \boldsymbol{a}_t, \boldsymbol{f}^*)) \right]$$

$$= \mathbb{E}_{\boldsymbol{\tau} \sim \hat{p}^{\boldsymbol{\pi}}(\boldsymbol{\tau})} \left[ \sum_{t \geq 0} \log \left( p(\boldsymbol{s}_{t+1}, r_t | \boldsymbol{s}_t, \boldsymbol{a}_t) \right) - \log(p(\boldsymbol{s}_{t+1}, r_t | \boldsymbol{s}_t, \boldsymbol{a}_t, \boldsymbol{f}^*)) \right]$$

$$= \mathbb{E}_{\boldsymbol{\tau} \sim \hat{p}^{\boldsymbol{\pi}}(\boldsymbol{\tau})} \left[ \sum_{t \geq 0} \log \left( p(\boldsymbol{s}_{t+1}, r_t | \boldsymbol{s}_t, \boldsymbol{a}_t) \right) - \log(p(\boldsymbol{w}_t)) \right]$$

$$= \mathbb{E}_{\boldsymbol{\tau} \sim \hat{p}^{\boldsymbol{\pi}}(\boldsymbol{\tau})} \left[ \sum_{t \geq 0} H(\boldsymbol{w}_t) - H(\boldsymbol{s}_{t+1}, r_t | \boldsymbol{s}_t, \boldsymbol{a}_t). \right]$$

$$= \mathbb{E}_{\boldsymbol{\tau} \sim \hat{p}^{\boldsymbol{\pi}}(\boldsymbol{\tau})} \left[ \sum_{t \geq 0} H(\boldsymbol{s}_{t+1}, r_t | \boldsymbol{s}_t, \boldsymbol{a}_t, \boldsymbol{f}) - H(\boldsymbol{s}_{t+1}, r_t | \boldsymbol{s}_t, \boldsymbol{a}_t). \right]$$

Here in the last step, we applied the equality that the entropy of $\boldsymbol{s}_{t+1}, r_t$ given the previous state, input, and dynamics is the same as the entropy of $\boldsymbol{w}_t$. In summary, we get

$$\mathbb{E}_{\boldsymbol{\tau} \sim \hat{p}^{\boldsymbol{\pi}}(\boldsymbol{\tau})} \left[ \sum_{t \geq 0} \log \left( E_{\boldsymbol{f}} \left[ p(\boldsymbol{s}_{t+1}, r_t | \boldsymbol{s}_t, \boldsymbol{a}_t, \boldsymbol{f}) \right] \right) - \log(p(\boldsymbol{s}_{t+1}, r_t | \boldsymbol{s}_t, \boldsymbol{a}_t, \boldsymbol{f}^*)) \right]$$

$$= \mathbb{E}_{\boldsymbol{\tau} \sim \hat{p}^{\boldsymbol{\pi}}(\boldsymbol{\tau})} \left[ \sum_{t \geq 0} -I(\boldsymbol{s}_{t+1}, r_t; \boldsymbol{f} | \boldsymbol{s}_t, \boldsymbol{a}_t) \right].$$

Therefore Eq. (18) can be rewritten as

$$\boldsymbol{\pi}^* = \arg \max_{\boldsymbol{\pi}} \mathbb{E}_{\boldsymbol{\tau} \sim \hat{p}^{\boldsymbol{\pi}}(\boldsymbol{\tau})} \left[ \sum_{t \geq 0} r_t - \alpha \log(\boldsymbol{\pi}(\boldsymbol{u}_t | \boldsymbol{z}_t)) + \alpha I(\boldsymbol{s}_{t+1}, r_t; \boldsymbol{f} | \boldsymbol{s}_t, \boldsymbol{a}_t) \right].$$

Intuitively, to bring our estimated distribution of the policy $\hat{p}^{\boldsymbol{\pi}}(\boldsymbol{\tau})$ closer to the desired distribution $p^*(\boldsymbol{\tau})$, we need to pick policies that maximize the underlying rewards while also gathering more information about the unknown elements of the MDP; the reward and dynamics model. While the former objective drives us in areas of high rewards under our estimated model, the latter objective ensures that our model gets more accurate over time.

## D ADDITIONAL EXPERIMENTS

We present additional experiments and ablations in this section.

**Covergence of intrinsic reward coefficient $\lambda$ for MAXINFOSAC**: In Fig. 8 we report the tuned intrinsic reward coefficient $\lambda$ for SACEIPO and MAXINFOSAC ($\alpha_2$ for MAXINFOSAC). As shown in Fig. 5 in Section 4, SACEIPO obtains suboptimal performance compared to MAXINFOSAC on the Pendulum, CartPole, and Walker tasks. We ascribe this to its tendency to underexplore, i.e. $\lambda \to 0$ much faster and converges to the local optima. Figure 8 validates our intuition since the intrinsic reward coefficient quickly decays to zero for SACEIPO whereas for MAXINFOSAC it does not.

**MAXINFORL with REDQ**: In Fig. 9 we combine MAXINFORL with REDQ. In all our tasks, REDQ does not obtain high rewards, we believe this is due to the high update to data ratio, which leads to the Q function converging to a sub-optimal solution. However, when combined with MAXINFORL, the resulting algorithm obtains considerably better performance and sample-efficiency.

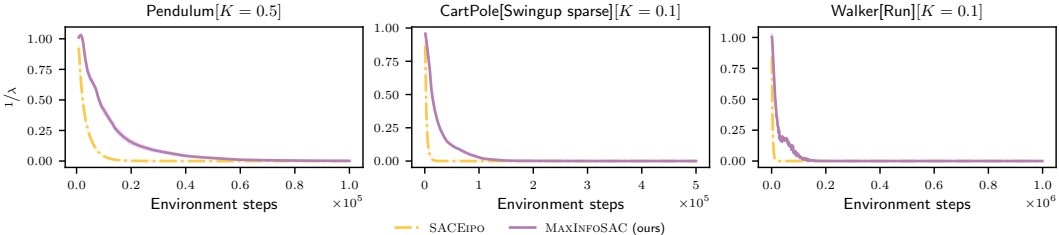

Figure 8: Evolution of the intrinsic reward coefficient $\lambda$ of SACEIPO and MAXINFOSAC over environment interaction.

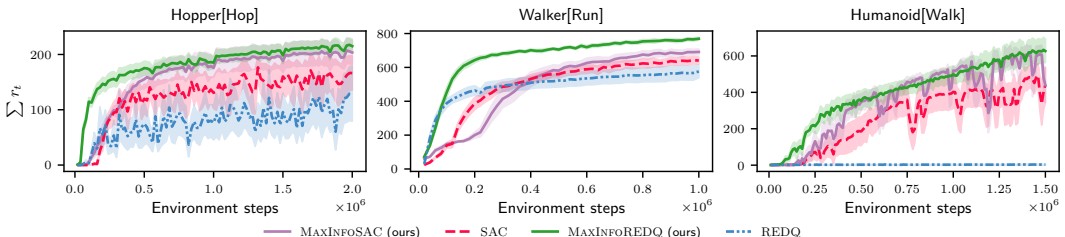

Figure 9: We combine MAXINFORL with REDQ (MAXINFOREDQ) and report the learning curves of SAC, REDQ, MAXINFOSAC, and MAXINFOREDQ.

**MAXINFORL with DrQv2 on additional DMC tasks**: In Fig. 10 we compare MAXINFODRQV2 with DrQv2 and MAXINFODRQ. For MAXINFODRQV2, we use a fixed noise variance with $\sigma = 0.2$ and $\sigma = 0.0$. We observe that on all tasks, MAXINFODRQV2 performs better than DrQv2, even when no noise is added for exploration. This empirically demonstrates that already by just maximizing the information gain, we get good exploration.

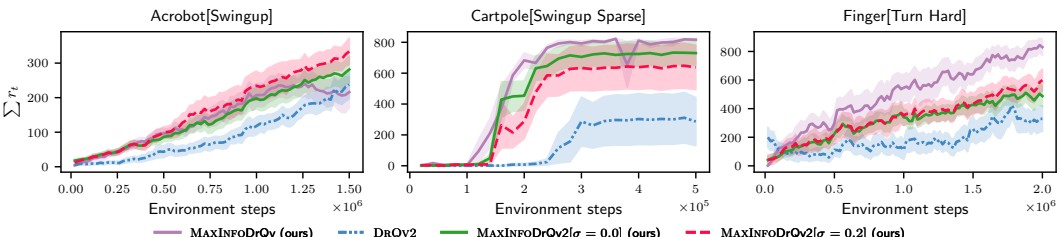

Figure 10: Learning curves of MAXINFODRQV2 with different noise levels $\sigma \in \{0.0, 0.2\}$ compared to DrQv2 and MAXINFODRQ.

In Fig. 11, we evaluate the exploration behavior of MAXINFODRQV2 without action noise on the DMC Humanoid Walk task. The figure shows that even without noise, MAXINFODRQV2 outperforms DrQv2, while the noise-free DrQv2 struggles to achieve high rewards.

**MAXINFOSAC with different intrinsic reward**: We evaluate MAXINFOSAC with RND as intrinsic reward. Moreover, we initialize a random NN model and train another model to predict the output of the random model as proposed in Burda et al. (2018). We train an ensemble of NNs to learn the random model and use their disagreement as the intrinsic reward[3]. Effectively, we replace the information gain term in Eq. (9) with the intrinsic reward derived from RND. In Fig. 12 we report the learning curves. From the figure, we conclude that MAXINFORL also works with RND as the exploration bonus. Moreover, while the standard MAXINFOSAC performs better on these tasks, MAXINFOSAC with RND performs significantly better than SAC. This illustrates the generality of our approach.

---

[3]Burda et al. (2018) use curiosity as the intrinsic reward but we observed that disagreement performed much better and robustly in our experiments.

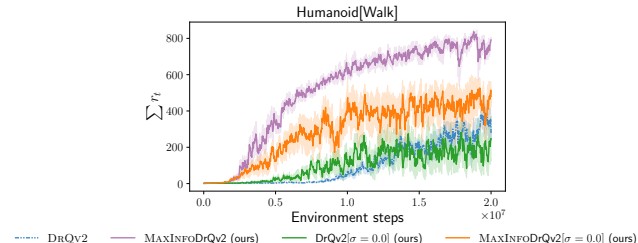

Figure 11: MAXINFODRQV2 evaluated on the humanoid walk task with no action noise.

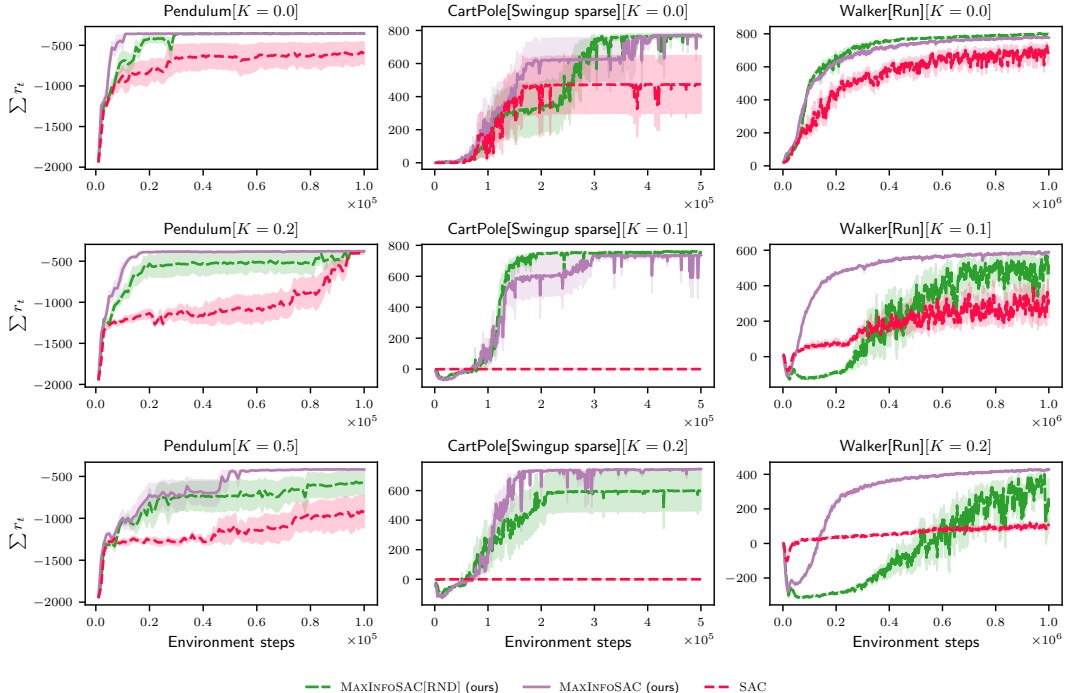

Figure 12: Learning curves of MAXINFOSAC with RND as the intrinsic reward, instead of the information gain, compared to SAC and standard MAXINFOSAC.

**Experiments with $\epsilon$–MAXINFORL** We also evaluate our $\epsilon$–greedy version from Section 3.1, $\epsilon$–MAXINFORL, empirically. For the intrinsic reward, we use the disagreement (Pathak et al., 2019) in the ensemble models of $f^*$ and as the base algorithm, we use SAC. In Fig. 13 we compare it to MAXINFOSAC and SAC and report the learning curves. We also evaluate $\epsilon$–MAXINFORL with different action costs, as in Section 4, in Fig. 14. For $\epsilon$ we specify a linear decay schedule (see Appendix E). In the figures, we observe that $\epsilon$–MAXINFORL performs better than SAC and often on par to MAXINFOSAC. However, $\epsilon$–MAXINFORL requires training two different actor-critic networks and manually specifying the schedule for $\epsilon$, as opposed to MAXINFOSAC which automatically trades off the extrinsic and intrinsic rewards.

**Experiments with $\epsilon$–MAXINFORL and Curiosity** In Fig. 15 we compare $\epsilon$–MAXINFORL with disagreement and curiosity as intrinsic rewards. We observe that overall both perform better than SAC, with disagreement obtaining slightly better performance than curiosity.

**Experiments with $\epsilon$–MAXINFORL with a switching frequency of one between policies.** In all the $\epsilon$–MAXINFORL experiments above, we switched between the extrinsic and intrinsic policies every 32 steps. Note that our proposed algorithm from Section 3.1 is still the same and we simply use a different schedule for the $\epsilon$ parameter. This is because we believe that instead of switching after every time step, following the extrinsic or intrinsic policies for several steps helps in exploration since we can collect longer trajectories of explorative data. In Fig. 16 we test this hypothesis on the pendulum environment, where we observe that indeed the switching frequency of 32 steps performs better.

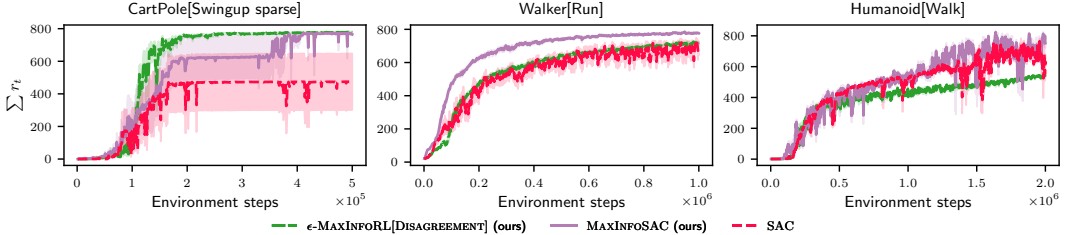

Figure 13: Learning curves of $\epsilon$–MAXINFORL with disagreement as the intrinsic reward, SAC and MAXINFOSAC.

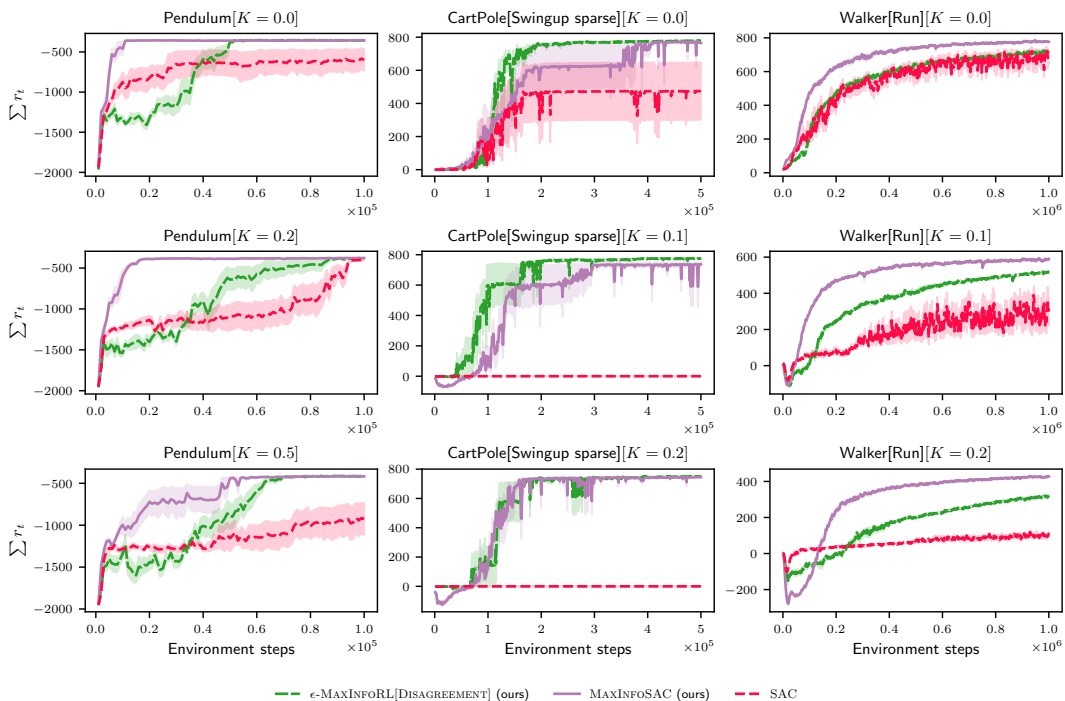

Figure 14: Learning curves of $\epsilon$–MAXINFORL with disagreement as the intrinsic reward, SAC and MAXINFOSAC for varying levels of action costs $K$.

**Experiments with OAC (Ciosek et al., 2019).** In Fig. 17, we compare OAC (Ciosek et al., 2019) an optimism based actor-critic algorithm with MAXINFOSAC. Furthermore, we also test a MAXINFORL version of OAC, called MAXINFOOAC. As depicted in the figure, we observe that while OAC performs better than SAC, the MAXINFORL perform the best.

**Experiments with DrM (Xu et al., 2024).** In Fig. 18, we evaluate DrM (Xu et al., 2024) another exploration algorithm for visual control tasks. DrM uses dormant ratio-guided exploration to solve challenging visual control problems. In our experiments, we evaluate the version of DrM without the expectile regression and use the same exploration schedule proposed in the paper. We combine DrM with MAXINFORL and report the performance on the humanoid and dog tasks from DMC.

## E EXPERIMENT DETAILS

In this section, we present the experiment details of MAXINFORL and our baselines.

**MAXINFOSAC** We train an ensemble of $P$ deterministic neural networks with mean squared error to predict the next state[4] and rewards. We evaluate the disagreement with $\boldsymbol{\sigma}_n(\boldsymbol{s}, \boldsymbol{a}) = \mathrm{Var}(\{\boldsymbol{f}_{\phi_i}(\boldsymbol{s}, \boldsymbol{a})\}_{i \in \{1,\ldots,P\}})$, where $\{\phi_i\}_{i \in \{1,\ldots,P\}}$ are the parameters of each ensemble member. We quantify the information gain using the upper bound in Equation (5) and for the aleatoric uncertainty,

---

[4]Specifically we predict $\boldsymbol{s}_{t+1} - \boldsymbol{s}_t$.

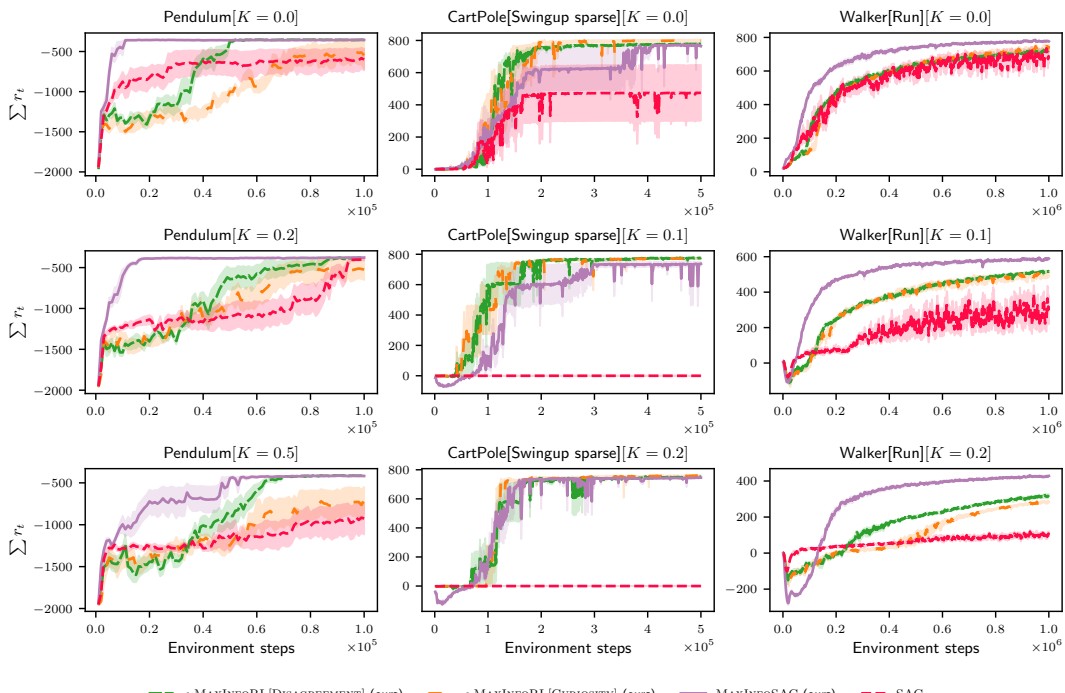

Figure 15: Learning curves of $\epsilon$–MAXINFORL with disagreement and curiosity as the intrinsic reward compared with SAC.

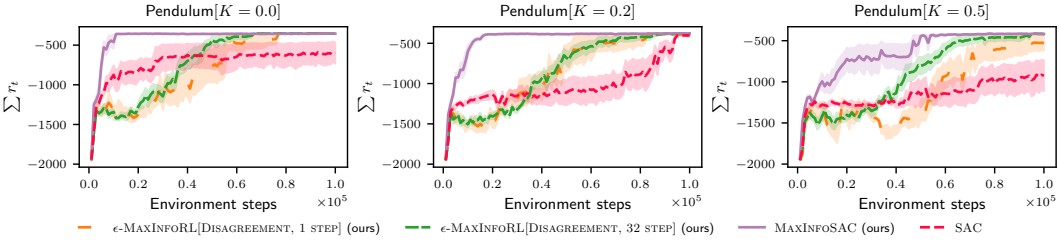

Figure 16: Learning curves of $\epsilon$–MAXINFORL with disagreement. We compare a version of $\epsilon$–MAXINFORL which switches between extrinsic and intrinsic policy every step with one which switches every 32 steps.

we use $\sigma = 10^{-3}$. In principle, we could learn the aleatoric uncertainty, however, we found that this approach already works robustly. Lastly, as suggested by Burda et al. (2018) we normalize the intrinsic reward/information gain term when training the critic.

A limitation of our work is that it requires training an ensemble of forward dynamics models to quantify the epistemic uncertainty. This is highlighted in Table 1 where we compare the training time required to train SAC with MAXINFOSAC. We report the numbers for our Stable-Baselines 3 (SB3) (Raffin et al., 2021) (torch-based) implementation and JaxRL (Kostrikov, 2021) (Jax-based) implementation.

Table 1: Computation cost comparison for MAXINFOSAC on NVIDIA GeForce RTX 2080 Ti

| Algorithm | Training time for 100k environment interactions |
|---|---|
| SAC (SB3) | 16.96 min +/- 1.64317 min |
| MAXINFOSAC (SB3) | 39 min +/- 1 min |
| SAC (JaxRL) | 5.6 min +/- 0.2min |
| MAXINFOSAC (JaxRL) | 7.3 min +/- 0.75 |

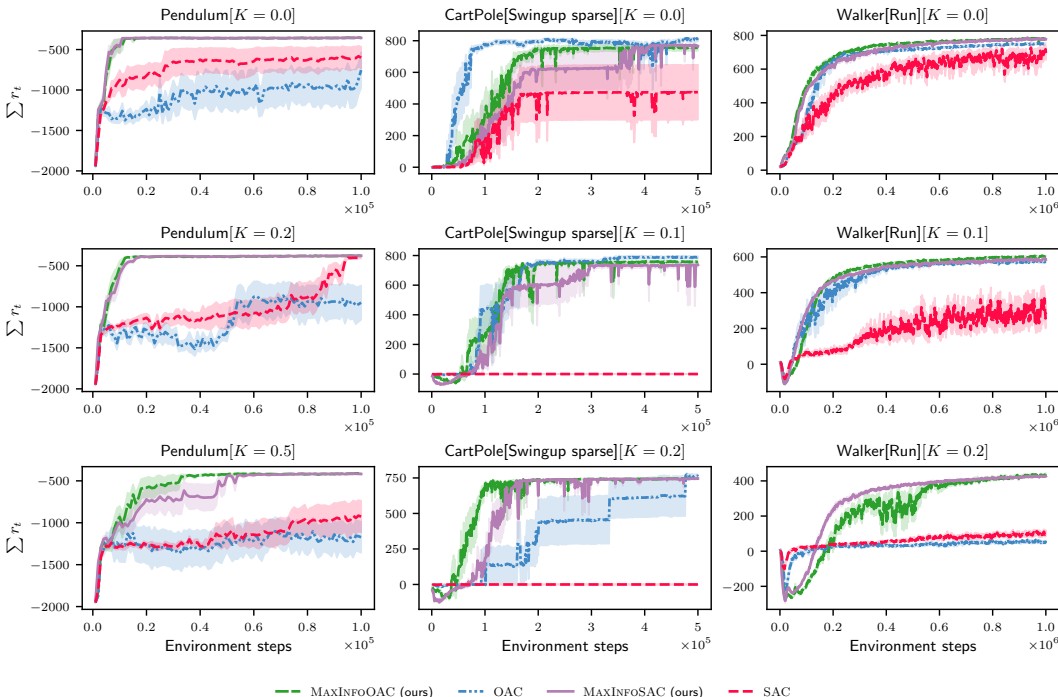

Figure 17: Learning curves of OAC compared with MAXINFOSAC and a MAXINFORL version of OAC (MAXINFOOAC).

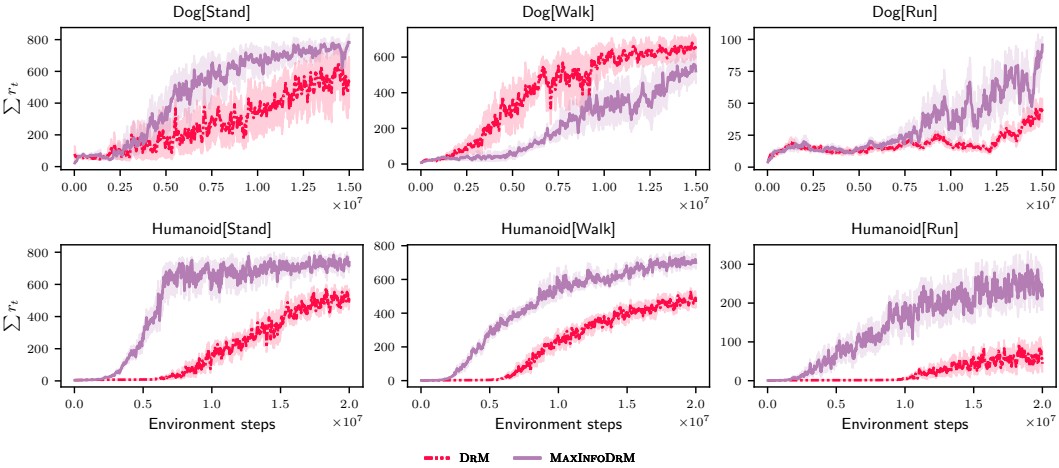

Figure 18: Learning curves of DrM compared with our version of DrM (MAXINFODRM).

As the number suggests MAXINFOSAC requires more time to train due to the additional computational overhead from learning the dynamics model. However, our JaxRL-based implementation already bridges the computational gap and is much faster. To further reduce the computational cost, we can consider cheaper methods for uncertainty quantification such as Osband et al. (2023).

**Explore then exploit** We train two different actor-critics, one for the extrinsic and one for the intrinsic reward. For the first 25% of the episodes, we only use the intrinsic actor for data collection. For the remaining 75%, the extrinsic actor is used. As the base actor-critic algorithm we use SAC. Both the intrinsic and extrinsic actor-critics share the same data buffer. Therefore, the data collected during the exploration phase is also used to train the policy for the exploitation phase. For the intrinsic reward, we use the disagreement calculated as specified above for MAXINFOSAC. We also eval-

uate this approach with curiosity as the intrinsic reward. Here we train a neural network model to predict the next state and reward and use the mean squared error in the prediction as the intrinsic reward.

**SACINTRINSIC** Here we follow the same procedure as proposed in Burda et al. (2018). We train two critics, one for the extrinsic reward and another one for the intrinsic rewards/exploration bonuses. We use the (normalized) information gain and policy entropy as the exploration bonuses. As suggested in Burda et al. (2018) the non-episodic returns are used in training the intrinsic critics. The policy is trained to maximize $\mathbb{E}_{\boldsymbol{s}\sim\mathcal{D}}[\mathbb{E}_{\boldsymbol{a}\sim\boldsymbol{\pi}(\cdot|\boldsymbol{s})}[Q_{\text{extrinsic}}^{\boldsymbol{\pi}_{\text{old}}}(\boldsymbol{s},\boldsymbol{a}) + Q_{\text{intrinsic}}^{\boldsymbol{\pi}_{\text{old}}}(\boldsymbol{s},\boldsymbol{a})]]$, where $\mathcal{D}$ is the data buffer.

**SACEIPO** We follow the same procedure as in SACINTRINSIC in training the critics. For the policy, we maximize a weighted sum of the extrinsic and intrinsic critics, i.e., the policy $\boldsymbol{\pi}$ is trained to maximize $\mathbb{E}_{\boldsymbol{s}\sim\mathcal{D}}[\mathbb{E}_{\boldsymbol{a}\sim\boldsymbol{\pi}(\cdot|\boldsymbol{s})}[\lambda Q_{\text{extrinsic}}^{\boldsymbol{\pi}_{\text{old}}}(\boldsymbol{s},\boldsymbol{a}) + Q_{\text{intrinsic}}^{\boldsymbol{\pi}_{\text{old}}}(\boldsymbol{s},\boldsymbol{a})]]$. We determine $\lambda$ using the extrinsic optimality constraint from Chen et al. (2021). For the policy $\boldsymbol{\pi}$, the extrinsic optimality constraint is given by

$$\mathbb{E}_{\boldsymbol{s}\sim\mathcal{D}}[\mathbb{E}_{\boldsymbol{a}\sim\boldsymbol{\pi}(\cdot|\boldsymbol{s})}[Q_{\text{extrinsic}}^{\boldsymbol{\pi}}(\boldsymbol{s},\boldsymbol{a})]] - \max_{\boldsymbol{\pi}'\in\Pi} \mathbb{E}_{\boldsymbol{s}\in\mathcal{D}}[\mathbb{E}_{\boldsymbol{a}\sim\boldsymbol{\pi}'(\cdot|\boldsymbol{s})}[Q_{\text{extrinsic}}^{\boldsymbol{\pi}'}(\boldsymbol{s},\boldsymbol{a})]] = 0$$

Effectively, we encourage the policy $\boldsymbol{\pi}$ that maximizes the intrinsic and extrinsic rewards performs at least as well as the policy which only maximizes the extrinsic ones. To evaluate this constraint, we train another actor and critic that greedily maximizes the extrinsic reward only, i.e., we use the data buffer to learn the solution $\pi_E^*$ for $\max_{\boldsymbol{\pi}'\in\Pi} \mathbb{E}_{\boldsymbol{s}\in\mathcal{D}}[\mathbb{E}_{\boldsymbol{a}\sim\boldsymbol{\pi}'(\cdot|\boldsymbol{s})}[Q_{\text{extrinsic}}^{\boldsymbol{\pi}'}(\boldsymbol{s},\boldsymbol{a})]]$.

$$L(\lambda) = \lambda \left( \mathbb{E}_{\boldsymbol{a}\sim\boldsymbol{\pi}(\cdot|\boldsymbol{s})}[Q_{\text{extrinsic}}^{\boldsymbol{\pi}}(\boldsymbol{s},\boldsymbol{a})]] - \mathbb{E}_{\boldsymbol{s}\sim\mathcal{D}}[\mathbb{E}_{\boldsymbol{s}\in\mathcal{D}}[\mathbb{E}_{\boldsymbol{a}\sim\pi_E^*(\cdot|\boldsymbol{s})}[Q_{\text{extrinsic}}^{\pi_E^*}(\boldsymbol{s},\boldsymbol{a})]]] \right)$$

Intuitively, if our constraint is not satisfied, i.e., $\mathbb{E}_{\boldsymbol{s}\sim\mathcal{D}}[\mathbb{E}_{\boldsymbol{s}\in\mathcal{D}}[\mathbb{E}_{\boldsymbol{a}\sim\pi_E^*(\cdot|\boldsymbol{s})}[Q_{\text{extrinsic}}^{\pi_E^*}(\boldsymbol{s},\boldsymbol{a})]]] > \mathbb{E}_{\boldsymbol{s}\sim\mathcal{D}}[\mathbb{E}_{\boldsymbol{a}\sim\boldsymbol{\pi}(\cdot|\boldsymbol{s})}[Q_{\text{extrinsic}}^{\boldsymbol{\pi}}(\boldsymbol{s},\boldsymbol{a})]]$, we increase $\lambda$, effectively focusing more on the extrinsic reward. We update $\lambda$ with gradient descent on $L(\lambda)$, akin to the temperature parameters in MAXINFOSAC.

**MAXINFODRQ** We add our information gain on top of DrQ using the same architecture and image augmentations from Yarats et al. (2021). The policy and critic updates are identical to MAXINFOSAC. However, since we deal with images, we train an ensemble of $P$ deterministic neural networks with mean squared error to predict the next state $\boldsymbol{s}_{t+1}$, rewards $r_t$, and a compressed image embedding $e_t$ which we obtain from the encoder for the observation $o_t$. Effectively, we compress the output of the encoder into a $d_e$ dimensional embedding using max pooling and use that as a label for our ensemble training. Crucially, we want the ensemble to learn and explore all unknown components of the MDP, the dynamics, rewards and observation model. For the state $\boldsymbol{s}$, we use the learned representation from the target critic.

**MAXINFODRQV2** We use the same procedure for training the ensemble model as MAXINFODRQ. For MAXINFODRQV2, we separate the intrinsic and extrinsic reward critic. This is primarily because DrQv2 uses $n$–step returns for training the extrinsic critic. By separating the two critics, we can train the extrinsic critic with the $n$–step returns and the intrinsic with the standard 1–step ones. This makes implementing the algorithm easier.

**MAXINFOSAC[RND]** In Fig. 12 we evaluate MAXINFOSAC with RND as intrinsic reward. We initialize a target neural network to predict an output embedding $\boldsymbol{y}$ given $(\boldsymbol{s},\boldsymbol{a})$. We train an ensemble of neural networks to learn the target network and use the disagreement as the intrinsic reward. The remaining training procedure is identical to MAXINFOSAC.

**$\epsilon$–MAXINFORL** We also evaluate the $\epsilon$–MAXINFORL algorithm from Equation (6). We train two actor-critics similar to the explore and then exploit baselines. However, instead of exploring for a fixed amount of iterations initially, we specify a probability $\epsilon_t$ to alternate between the exploitation and exploration phases. Furthermore, changing the between policies after every time-step can lead to noisy trajectories. Therefore, we choose a frequency $f = 32$ for the switches. That is, we change the sampling strategy (explore or exploit) after a minimum of $f\,steps$.

### E.1 ALGORITHMS

We provide more details about MAXINFOSAC, MAXINFODRQ, and MAXINFODRQV2. All three are off-policy actor-critic algorithms and learn two critics $Q_{\theta_1}$ and $Q_{\theta_2}$ to avoid overestimation bias. From here on, we denote with $\psi$ the parameters of the policy and $\bar{\theta}_{1,2}, \bar{\psi}$ the parameters of the target critics and policy.

**Losses for MAXINFOSAC and MAXINFODRQ**   We sample a trajectory $\boldsymbol{\tau}$ from $\mathcal{D}$ and use the following for the critic and policy losses.

Critic loss:

$$J_Q(\theta) = \frac{1}{2}\mathbb{E}_{\boldsymbol{\tau}\sim\mathcal{D}}[(Q_\theta(\boldsymbol{s},\boldsymbol{a}) - y)^2]$$

$$y = r + \gamma \min_{k\in\{1,2\}} Q_{\bar{\theta}_k}(\boldsymbol{s}',\boldsymbol{a}') - \alpha_1 \log \boldsymbol{\pi}_\psi(\cdot|\boldsymbol{s}') + \alpha_2 I_u(\boldsymbol{s}',\boldsymbol{a}'). \tag{19}$$

where $\boldsymbol{a}' \sim \boldsymbol{\pi}_\psi(\cdot|\boldsymbol{s}')$.

Policy loss:

$$J_{\boldsymbol{\pi}}(\psi) = \mathbb{E}_{\boldsymbol{s}\sim\mathcal{D}}[\mathbb{E}_{\boldsymbol{a}\sim\boldsymbol{\pi}_\psi}[\alpha_1 \log \boldsymbol{\pi}_\psi(\cdot|\boldsymbol{s}) - \alpha_2 I_u(\boldsymbol{s},\boldsymbol{a}) - \min_{k\in\{1,2\}} Q_{\theta_k}(\boldsymbol{s},\boldsymbol{a})]]. \tag{20}$$

Entropy coefficient loss;

$$J(\alpha_1) = \mathbb{E}_{\boldsymbol{s}\sim\mathcal{D}}[-\alpha_1(\log \boldsymbol{\pi}_\psi(\cdot|\boldsymbol{s}) + \bar{H})]. \tag{21}$$

**Critic loss for MAXINFODRQV2**   For MAXINFODRQV2 we train separate critics for the intrinsic and extrinsic rewards. This allows us to use $n$–step returns for the extrinsic rewards while keeping a simple implementation, using 1–step returns, for the intrinsic ones. Effectively, for computational efficiency, we adapt the critic network to output two heads, one for the extrinsic and one for the intrinsic reward.

Critic loss:

$$J_Q(\theta) = \frac{1}{2}\mathbb{E}_{\boldsymbol{\tau}\sim\mathcal{D}}[(Q_\theta^{\text{extrinsic}}(\boldsymbol{s},\boldsymbol{a}) - y^{\text{extrinsic}})^2 + (Q_\theta^{\text{intrinsic}}(\boldsymbol{s},\boldsymbol{a}) - y^{\text{intrinsic}})^2]$$

$$y^{\text{extrinsic}} = \sum_{l=0}^{n} \gamma^l r_l + \gamma^n \min_{k\in\{1,2\}} Q_{\bar{\theta}_k}^{\text{extrinsic}}(\boldsymbol{s}_n,\boldsymbol{a}_n)$$

$$y^{\text{intrinsic}} = I_u(\boldsymbol{s},\boldsymbol{a}) + \gamma \min_{k\in\{1,2\}} Q_{\bar{\theta}_k}^{\text{intrinsic}}(\boldsymbol{s}',\boldsymbol{a}'). \tag{22}$$

where $\boldsymbol{a}_n \sim \boldsymbol{\pi}_\psi(\cdot|\boldsymbol{s}_n)$ and $\boldsymbol{a}' \sim \boldsymbol{\pi}_\psi(\cdot|\boldsymbol{s}')$.

Policy loss:

$$J_{\boldsymbol{\pi}}(\psi) = \mathbb{E}_{\boldsymbol{s}\sim\mathcal{D}}\left[\mathbb{E}_{\boldsymbol{a}\sim\boldsymbol{\pi}_\psi}\left[-\left(\min_{k\in\{1,2\}} Q_{\theta_k}^{\text{extrinsic}}(\boldsymbol{s},\boldsymbol{a}) + \alpha_2 \min_{k\in\{1,2\}} Q_{\theta_k}^{\text{intrinsic}}(\boldsymbol{s},\boldsymbol{a})\right)\right]\right]. \tag{23}$$

Loss for the information gain coefficient

$$J(\alpha_2) = \alpha_2 \left(\mathbb{E}_{\boldsymbol{s}\sim\mathcal{D}}[\mathbb{E}_{\boldsymbol{a}\sim\boldsymbol{\pi}_\psi}[I_u(\boldsymbol{s},\boldsymbol{a})] - \mathbb{E}_{\bar{\boldsymbol{a}}\sim\bar{\boldsymbol{\pi}}_\psi}[I_u(\boldsymbol{s},\bar{\boldsymbol{a}})]]\right). \tag{24}$$

Loss of ensemble model for MAXINFOSAC. Let $\Phi = \{\phi_i\}_{i\in\{1,...P\}}$

$$J(\Phi) = \mathbb{E}_{\boldsymbol{\tau}\sim\mathcal{D}}\left[\sum_{i=1}^{P} \|\boldsymbol{f}_{\phi_i}(\boldsymbol{s},\boldsymbol{a}) - y\|^2\right] \tag{25}$$

$$y = \left[(\boldsymbol{s}' - \boldsymbol{s})^\top, r\right]. \tag{26}$$

Loss of ensemble model for MAXINFODRQ and MAXINFODRQV2.

$$J(\Phi) = \mathbb{E}_{\boldsymbol{\tau}\sim\mathcal{D}}\left[\sum_{i=1}^{P} \|\boldsymbol{f}_{\phi_i}(\boldsymbol{s},\boldsymbol{a}) - y\|^2\right] \tag{27}$$

$$y = \left[\boldsymbol{s}'^\top, r, e^\top\right]. \tag{28}$$

In Algorithm 1 we present the main structure of the algorithms. We omit algorithm-specific details of DrQ, DrQv2 such as noise scheduling and image augmentation in Algorithm 1 and refer the reader to the respective papers for those details (Yarats et al., 2021; 2022).

---

**Algorithm 1** Algorithm structure for MAXINFOSAC, MAXINFODRQ, MAXINFODRQV2

---

**Init:** $\theta_1, \theta_2, \psi$
$\bar{\theta}_1 \leftarrow \theta_1, \bar{\theta}_2 \leftarrow \theta_2, \bar{\psi} \leftarrow \psi$                      ➤ Initialize target
$\mathcal{D} = \emptyset$
**for** iterations $n = 1, \dots, n$ **do**
    **for** each environment step **do**
        $\boldsymbol{a}_t \sim \boldsymbol{\pi}_\psi(\cdot|\boldsymbol{s}_t)$                              ➤ Sample action
        $\boldsymbol{s}_{t+1}, r_t \sim p(\boldsymbol{s}_{t+1}, r_t|\boldsymbol{s}_t, \boldsymbol{a}_t)$              ➤ Observe state and reward
        $\mathcal{D} \leftarrow \mathcal{D} \cup \{\boldsymbol{s}_t, \boldsymbol{a}_t, \boldsymbol{s}_{t+1}, \boldsymbol{a}_t\}$
    **end for**
    **for** each gradient step **do**
        Update critics with stochastic gradient descent (SGD) on $J_Q(\boldsymbol{\theta}_1) + J_Q(\boldsymbol{\theta}_2)$
        Update critics with SGD on $J_{\boldsymbol{\pi}}(\psi)$
        Update temperatures with SGD on $J(\alpha_1), J(\alpha_2)^5$
        Update ensemble with SGD on $J(\Phi)$.
        Update $\bar{\theta}_{1,2}, \psi$                           ➤ Policy update target updates
    **end for**
**end for**

---

## E.2 HYPERPARAMETERS

Table 2: Hyperparameters for results in Section 4.

| Environment | Learning rate | Policy/Critic Architecture | Model architcture | Polyak Coefficient | action repeat | feature and embedding dim |
|---|---|---|---|---|---|---|
| State Based Tasks | | | | | | |
| DMC and Gym | $3 \times 10^{-4}$ | (256, 256) | | 0.005 | 1 for Gym 2 for DMC | - |
| Humanoid Bench | $5 \times 10^{-5}$ | | | 0.005 | | - |
| Visual Control tasks | | | | | | |
| DMC | $3 \times 10^{-4}$ for MAXINFODRQ, else $10^{-4}$ $10^{-4}$ for the model and $\alpha_2$ | (256, 256) | (256, 256) | 0.005 for MAXINFODRQ, else $10^{-4}$ | 2 | Feature: 50 Embedding: 32 |
| Humanoid and Dog tasks (DrM) | $8 \times 10^{-5}, 8 \times 10^{-5}$ for the model and $\alpha_2$ | (1024, 1024) | (1024, 1024) | 0.01 | 2 | Feature: 100 Embedding: 32 |

