# OpenReview forum: "MaxInfoRL: Boosting exploration in reinforcement learning through information gain maximization"
_ICLR.cc/2025/Conference — ICLR 2025 Poster_

### Official Review · Reviewer_XVpz · 2024-10-27

**Soundness:** 3
**Presentation:** 3
**Contribution:** 3
**Rating:** 8
**Confidence:** 2

**Summary:**

The paper, titled "MAXINFORL: Boosting Exploration in Reinforcement Learning through Information Gain Maximization," introduces a new approach to exploration in reinforcement learning by focusing on maximizing information gain. The authors propose the MAXINFORL framework, which augments standard exploration techniques, like Boltzmann and ε-greedy, with intrinsic rewards based on information gain to direct exploration toward more informative transitions.

Key contributions include a method that integrates intrinsic and extrinsic exploration objectives seamlessly, sublinear regret performance in multi-armed bandit settings, and the use of ensemble dynamics models to estimate information gain. The authors validate MAXINFORL across standard benchmarks in state-based and visual control tasks, showing significant performance improvements over baseline algorithms like SAC and DrQ. The approach demonstrates particularly strong results in complex exploration scenarios, such as those with sparse rewards and high-dimensional visual control tasks, indicating MAXINFORL’s potential for enhancing sample efficiency and performance in hard-to-solve RL tasks.

**Strengths:**

1) **Significance of the Problem**: The paper addresses an essential challenge in reinforcement learning — achieving efficient and directed exploration in environments where rewards are sparse or non-informative. By focusing on maximizing information gain, the proposed method has potential implications for enhancing the performance of RL agents in complex and high-dimensional tasks, which makes the work particularly impactful.

2) **Clear and Well-Organized Structure**: The paper is well-structured, providing a logical progression from the problem introduction through the methodology and experimental validation. The explanations of both theoretical insights and practical implementation are clear, making it easy for readers to understand how the proposed MAXINFORL framework improves upon existing exploration methods.

3) **Robust Experimental Results**: The experimental results across various benchmarks demonstrate that MAXINFORL consistently outperforms baseline methods in terms of sample efficiency and exploration effectiveness. The use of challenging tasks, including both state-based and visual control problems, further strengthens the validity and robustness of the presented results.

4) **Effective and Intuitive Methodology**: The MAXINFORL framework combines intrinsic rewards, specifically information gain, with established exploration strategies like Boltzmann exploration in a theoretically sound and computationally feasible way. The use of ensemble dynamics to estimate information gain and auto-tuning mechanisms to balance exploration objectives adds to the approach's practicality, making it a sensible and valuable contribution to RL research.

**Weaknesses:**

1) **Clarity on the Importance of Information Gain**: The core insight of using information gain as an intrinsic reward requires further clarification. While the paper emphasizes its role in enhancing exploration, a more illustrative example could demonstrate its unique contribution compared to other intrinsic rewards. This would help readers appreciate the specific novelty and advantage of information gain within this framework.

2) **Limited Novelty in the Trade-off Mechanism**: The method for balancing intrinsic and extrinsic rewards, while effective, lacks substantial novelty. The approach builds on existing auto-tuning and temperature parameter techniques without introducing significant new strategies for trade-off optimization. An exploration of alternative or novel mechanisms for managing this balance, particularly in complex environments, would strengthen the paper’s contribution and originality.

**Questions:**

please see weaknesses.

---

> ### Author Response · Authors · 2024-11-17
> **Response to Reviewer XVpz**
>
> We thank the reviewer for their feedback and for acknowledging the significance of our work, the empirical evaluation, and our intuitive methodology. Below we address the concerns raised by the reviewer.
>
> **Weaknesses**
>
> **Clarity on the Importance of Information Gain**:  As we discuss in Section 2.4, information gain is commonly used for exploration for experiment design in bandits and RL literature. In equation (8) in the paper, we show that our approach also results in maximizing entropy in both the state and action space as opposed to just the action space which is what standard Boltzmann exploration does. In Figure 4, we empirically show this on the Pendulum, where we plot the states visited during exploration by MaxInfoRL and compare it to SAC. Lastly, in Figures 12 and 15, we combine our method with different intrinsic rewards and show that generally the information gain/model uncertainty performs better. However, we are happy to consider additional illustrations that the reviewer thinks might add more to the clarity of the reader.
>
>
> **Limited Novelty in the Trade-off Mechanism**: We agree that our trade-off mechanism builds on existing strategies in the RL literature. However, we think this is a strength of our work and part of the reason why our approach works so robustly across different benchmarks and tasks. In particular, these auto-tuning mechanisms are widely applied for the action entropy (SAC, DrQ, REDQ, DrO etc., all use this mechanism and have achieved strong empirical performance). More importantly, they are simple to integrate into an RL algorithm, which in itself is an important attribute for RL practitioners. Therefore, we specifically catered our approach to benefit from these existing mechanisms and thus make it flexible, robust, and also easily accessible for RL researchers. To the best of our knowledge, we are the first to establish this link between intrinsic exploration and the trade-off mechanisms discussed above.
>
> We once again thank the reviewer for their active engagement in the review process and hope our responses enhance the confidence of the reviewer about our work and would be happy if they prompt the reviewer to improve our score

---

> > ### Author Response · Authors · 2024-11-20
> > **Follow up on response to Reviewer XVpz**
> >
> > Dear Reviewer,
> >
> > Having addressed your concerns, we would appreciate your feedback during this discussion period. Let us know if there are any further questions that we can clarify, otherwise, we would appreciate it if you would consider increasing your score.
> >
> > Thanks!

---

> > ### Comment · Reviewer_XVpz · 2024-11-22
> >
> > Thanks for your feedback. My concerns are mostly addressed. I will improve my score to 7.

---

> > > ### Author Response · Authors · 2024-11-23
> > >
> > > We thank the reviewer for reevaluating and increasing our score. We are happy to hear that our rebuttal could address their concerns.

---

### Official Review · Reviewer_8tDw · 2024-10-29

**Soundness:** 3
**Presentation:** 3
**Contribution:** 2
**Rating:** 6
**Confidence:** 4

**Summary:**

The paper tackles the issue of poor exploration techniques based on dithering (like $\epsilon$-greedy and Boltzmann exploration) that are often used in continuous control algorithms such as SAC and REDQ by adapting them to include an intrinsic reward bonus based on the information gain from observing a particular transition.

**Strengths:**

The empirical evaluation is extensive in terms of the domains evaluated on.

The theoretical result in Theorem 3.1 demonstrating convergence of their method is important and I'm pleased the authors included it.

I think the paper is generally well written and easy to follow.

**Weaknesses:**

There is significant related work missing from the paper as there is no discussion of the rich literature on Bayesian RL (in particular, methods that achieve deep exploration [2]), which have been tackling the same problem highlighted by the authors since at least 2018.

I also have concerns with both the empirical and theoretical justification of the algorithm

Theoretical justification of algorithm:

Issue I: Regret bound is not particularly strong

The authors derive a regret bound in the bandit setting of the order $\mathcal{O}(T^{1-\alpha}\log(T)^k + T^{2\alpha})$ for $\alpha\in(0.25,0.5)$ and some $k$ depending on the GP kernel. Whilst this achieves sub-linear regret in the Bandit setting, this is not a high bar to clear and there are many approaches that achieve much better regret bounds like UCB and Thompson sampling, which achieves Lai and Robbins lower bound of $\log(T)$.

Issue II: Analysis in the bandit setting

Carrying out regret analysis in the bandit setting is not sufficient to make claims about regret in the full MDP setting. For example, as discussed already, it is known that Thompson sampling achieves Lai and Robbins lower bound of $\log(T)$ in the bandit setting, but posterior sampling does not achieve optimal regret in the full MDP setting [1].

For this reason, I'm not convinced there is a strong theoretical case for the algorithm

Empirical Justification:

The authors focus on comparing against algorithms that don't have particularly strong exploration properties and this feels a bit like a straw man to me. Given the focus on exploration, I'd expect an empirical evaluation that compares to, for example, [2][3][4][5] to name a few, which all have better theoretical exploration properties and are known to perform well on challenging exploration continuous control tasks. As an example, an empirical evaluation similar to [6] which compares against state of the art exploration methods would be the evaluation I'd expect from a paper on exploration in model-free continuous control.

For this reason, I don't think there is a strong empirical case for the algorithm.

[1] Buening et al. Minimax-Bayes Reinforcement Learning, 2023 https://arxiv.org/pdf/2302.10831

[2] Osband et al. Randomized prior functions for deep reinforcement learning, 2018 https://arxiv.org/pdf/1806.03335

[3] Fellows et al. Bayesian Bellman Operators, 2021 https://arxiv.org/pdf/2106.05012

[4] Fellows et al. Bayesian Exploration Networks, 2023 https://arxiv.org/pdf/2308.13049

[5] Coisek et al. Better Exploration with Optimistic Actor-Critic, 2019 https://arxiv.org/pdf/1910.12807

[6]  Tasdighi et al. Deep exploration with PAC-Bayes, 2024 https://arxiv.org/pdf/2402.03055

**Questions:**

Can the authors prove a better regret bound for their approach that rivals approaches like posterior sampling?
Is there a reason to neglect the algorithms highlighted in [6] in favour of their approach?

---

> ### Author Response · Authors · 2024-11-17
> **Response to Reviewer 8tDw**
>
> We thank the reviewer for their feedback and for acknowledging our thorough empirical evaluation. Below we provide a detailed response to the questions raised by the reviewer.
>
> ***Weaknesses***
>
> **Regret bound**: We provide an upper bound for the regret and we think comparing a loose lower bound with the upper bound is not informative since the two do not match. To the best of our knowledge, the regret bound even for UCB or TS approaches with GPs, i.e., continuous input and output spaces, scales with $\beta_T \sqrt{\gamma_T T}$ and [1] also provide a tighter lower bound that scales similar to the upper one, i.e., is not $log(T)$. Moreover, it is widely believed that the minimax optimal rate is polynomial (see https://proceedings.mlr.press/v178/open-problem-vakili22a/open-problem-vakili22a.pdf and references therein).
> However, we acknowledge that our upper bound is weaker than TS and UCB methods. This is not surprising as $\epsilon$--greedy generally has a weaker regret bound than UCB-like algorithms such as UCRL [2, 8, 9]. Yet, these methods are widely applied in deep RL due to their simplicity and scalability.
>
> **Regret analysis in the bandit setting**: We agree that it is not straightforward to claim sublinear regret in the MDP setting with analysis for the bandit case. However, our intention for the bandit analysis is not to make this claim. Rather we used the bandit analysis as inspiration for our deep RL algorithm. This is partially motivated by looking at other works such as [2]. To further clarify this, we changed the line 210 “In Appendix A, we provide a theoretical justification for our approach and study $\epsilon$–MAXINFORL in the simplified setting of multi-armed bandit (MAB).” to “In Section A, to give a theoretical intuition of our approach, we study $\epsilon$-MAXINFORL in the simplified setting of multi-armed bandit (MAB).” In addition, while we think deriving a regret bound for our algorithm is an interesting direction for future work, we focus the scope of this paper more on the wide applicability of our approach, which is why we evaluate it across several standard deep RL benchmarks on both state-based and visual control tasks, as also acknowledged by the reviewer.
>
> **Choice of baselines**: The main objective of our method is to boost existing RL algorithms by combining intrinsic exploration and extrinsic rewards in a principled manner. Our work allows us to extend several RL algorithms, such as SAC, DrQ, DrQv2, benefiting from their simplicity while also having principled exploration from intrinsic reward which are widely studied in the RL literature.
> Therefore, besides considering the naive exploration algorithms such as SAC, we use intrinsic exploration baselines such as curiosity and disagreement for our comparison. In addition, we also compare our algorithm with [3, 4] which also try to combine extrinsic and intrinsic rewards. To the best of our knowledge, these are state-of-the-art approaches for combining intrinsic and extrinsic exploration.
> In addition, we also compare our method to REDQ (see Figure 9) which is one of the most competitive baselines in the paper [5] proposed by the reviewer. However, following the reviewer’s feedback we have also added experiments with the algorithm OAC [6] and our MaxInfoRL version of OAC to the paper (see Figure 17). We chose OAC as we think it fits our set of baselines the best since it is also a soft-actor critic algorithm that is based on principled exploration via optimism. Furthermore, we could find an official open-source implementation of the algorithm as a reference.
>
> For our visual control tasks, we compare with DrQv2 and now DrM (Figure 18) which are SOTA model-free RL algorithms known to solve the challenging problem of humanoid control from visual observations [7]. We show that our method outperforms it. Given our collection of baselines, and now OAC in addition, we believe our empirical evaluation makes a strong case for our method, as also acknowledged by the other reviewers.
>
> **Related Works**: Thanks for listing the related work. We will include them in the revised/final version of our paper. The provided references are interesting and relevant for exploration in RL. However, as we mentioned in the previous point the goal of our work is to combine extrinsic and intrinsic rewards in a principled manner, which results in a simple, scalable and flexible algorithm, which is what we also show in our experiments where we thoroughly evaluate our approach on state-based and visual-control benchmarks while also combining it with several model-free RL base algorithms such as SAC, REDQ, DrQv2 etc.

---

> > ### Author Response · Authors · 2024-11-17
> > **Response to Reviewer 8tDw continued**
> >
> > ***Questions***
> >
> > **Regret bound**: See our response to the first two points in the weaknesses section.
> >
> > **Reason for applying our approach**: As highlighted in the third point in the weaknesses section, the focus of our work is to combine intrinsic exploration with extrinsic rewards. Note that both intrinsic exploration and naive exploration algorithms such as SAC, DrQ, and DrQv2 are commonly used in RL. We believe part of the reason why this is the case is because these algorithms are fairly simple and also scale to higher dimensional settings such as visual control. By combining the two approaches, our framework boosts naive exploration methods while enjoying the exploration performance of intrinsic exploration approaches. Moreover, our framework is flexible and can be applied to several model-free algorithms and different intrinsic rewards – we apply it to SAC, REDQ, DrQ, DrQv2, and now OAC and DrM [9] to make this point, while also testing our method with RND and curiosity as intrinsic rewards. This flexibility makes our work much more general than individual algorithms such as REDQ. Furthermore, as we show in our experiments, combining these methods with our approach improves the performance.
> > Moreover, our approach achieves this improvement while enjoying the simplicity of naive exploration schemes, i.e., the training of the base algorithm SAC, DrQ, and etc remains the same and we simply add an additional (intrinsic) reward with the appropriate, auto-tuned scale during training. Therefore, we think the paper addresses an important problem of combining extrinsic and intrinsic rewards (also acknowledged by reviewers bvBz and XVpz) and provides a simple, flexible, and scalable solution. We empirically show the performance gain of our approach across several benchmarks on both state-based and visual-control tasks. We believe this makes our algorithm impactful and useful for many RL practitioners.
> >
> > We thank the reviewer for their active engagement in the review process and hope our responses address the reviewer’s main concern and highlight the key challenge our work aims to tackle better. We’d appreciate it if the reviewer would consider increasing our score and we are happy to answer any further questions.
> >
> > **References**:
> > 1. Scarlett, Jonathan, Ilija Bogunovic, and Volkan Cevher. "Lower bounds on regret for noisy gaussian process bandit optimization." Conference on Learning Theory. PMLR, 2017.
> > 2. Cesa-Bianchi, Nicolò, et al. "Boltzmann exploration done right." Advances in neural information processing systems 30 (2017).
> > 3. Chen, Eric, et al. "Redeeming intrinsic rewards via constrained optimization." Advances in Neural Information Processing Systems 35 (2022)
> > 4. Burda, Yuri, et al. "Exploration by random network distillation." arXiv preprint arXiv:1810.12894 (2018).
> > 5. Tasdighi et al. Deep exploration with PAC-Bayes, 2024 https://arxiv.org/pdf/2402.03055
> > 6. Ciosek, Kamil, et al. "Better exploration with optimistic actor critic." Advances in Neural Information Processing Systems 32 (2019)
> > 7. Yarats, Denis, et al. "Mastering visual continuous control: Improved data-augmented reinforcement learning." ICLR (2022)
> > 8. S. P. Singh, T. Jaakkola, M. L. Littman, and Cs. Szepesvári. Convergence results for single-step on-policy reinforcement-learning algorithms. Machine Learning, 38(3):287–308, 2000.
> > 9. Xu, Guowei, et al. "Drm: Mastering visual reinforcement learning through dormant ratio minimization." ICLR (2024).

---

> > > ### Author Response · Authors · 2024-11-20
> > > **Follow up on response to Reviewer 8tDw**
> > >
> > > Dear Reviewer,
> > >
> > > Having addressed your concerns, we would appreciate your feedback during this discussion period. Let us know if there are any further questions that we can clarify, otherwise, we would appreciate it if you would consider increasing your score.
> > >
> > > Thanks!

---

> > > ### Comment · Reviewer_8tDw · 2024-11-21
> > >
> > > I thank the authors for the clarification. Having read their response, I maintain that the theoretical justification is not strong enough as I would like to see a derivation in the full MDP setting with comparison to existing results on SOTA Bayesian methods.
> > >
> > > I appreciate the authors including OAC in their experiments. I would have preferred a comparison to a more SOTA Bayesian method, but provided adequate discussion of these approaches are provided. One thing, whilst I concede no Bayesian method is a simple as MaxInfoRL, I think it is misleading to say Bayesian methods aren't scalable and flexible algorithm. The linked papers show methods scale to the same domains (and harder) considered here. By definition, Bayesian RL (BRL) is more flexible than frequentist heuristics as exploration is inherent in the BRL objective, regardless of the MDP. In contrast, frequentist methods need heuristics that are tailored to specific domains. SAC and PPO for example fail miserably on a simple MountainCar problem, whereas Bayesian methods can solve these easily.
> > >
> > > Regardless, I appreciate there is some value to MaxInfoRL purely from the perspective of a practitioner. I will raise my score to 6 accordingly.

---

> > > > ### Author Response · Authors · 2024-11-21
> > > > **Response to Reviewer 8tDw continued**
> > > >
> > > > Dear Reviewer,
> > > >
> > > > Thanks for the active engagement in the rebuttal period, your feedback, and for increasing your score. Below is our response.
> > > >
> > > > **Theory**: As we mentioned in our response above, our regret bound is to give an intuition for the approach and derive the deep RL algorithm. Therefore, it is not a theoretical guarantee for the RL algorithm itself. We have already updated the paper to clarify this further. Furthermore, we agree that giving a regret for the full MDP setting is an interesting direction for future work, and will point this out in the paper as well.
> > > >
> > > > **Using OAC as baseline**:  In the most recent BRL paper referred by the reviewer (reference 6 listed by the reviewer which is also concurrent ICLR submission: https://openreview.net/forum?id=n4HH7g9hxk), we observe that REDQ is mostly on par with the BRL baselines, and we are indeed comparing with REDQ in our paper. Furthermore, in our rebuttal, we highlight our reasons for using OAC. In particular, we could find an official open-source codebase for the OAC algorithm for reference, which we could not for the BRL algorithms mentioned by the reviewer.
> > > >
> > > > **Scalability**: With scalability, we meant that our method can solve high-dimensional tasks such as visual control for the humanoid or dog tasks from DMC. These tasks are known to be very challenging to solve in the deep RL community (see references 7, and 9) and we have not seen the Bayesian RL methods discussed during the rebuttal applied to these settings.
> > > >
> > > > **Flexibility**:  With flexibility, we mean that our approach can be combined with any base RL algorithm as an add-on introduced to boost the existing method. Furthermore, while we agree that methods such as SAC can fail on the simplest tasks such as the MountainCar, from our experiments we observe that MaxInfoRL doesn’t. In particular, see Figure 2, where we show that boosting SAC with our framework solves the MountainCar task.
> > > >
> > > >
> > > > Hence, given the simplicity of MaxInfoRL (also acknowledged by the reviewer), the scalability to challenging tasks such as the visual control ones discussed above, and the flexibility to combine our approach with any base RL algorithm without requiring any additional hyperparameter tuning, we think MaxInfoRL is a very strong (SOTA on visual control benchmarks) and practical class of RL algorithms.

---

> ### Comment · Reviewer_8tDw · 2024-11-22
>
> I think [6] is a great example to rebut the authors' points.
>
> The authors of [6] implemented BEN from scratch (the original paper is for discrete domains, this work extends it to continuous). There is at least one BootPriorDQN+ implementation on Github that I found with a quick Google https://github.com/johannah/bootstrap_dqn. From past experience, it's very easy to reach out to authors to get implementations as most are willing to share them.
>
> Figure 4 from [6] shows Bayesian methods can solve incredibly challenging high dimensional domains. Most impressive is sparse humanoid. I've never seen another approach do this before. Moreover, Verse Sparse Ant can be solved by several methods. The only sparse environment I can find in the authors' paper is Cartpole, which from experience is relatively simple.
>
> Most Bayesian methods can be added on top of existing algorithms as they just affect exploration. I have added Bayesian exploration to many approaches, including PPO, DQN, RedQ etc.

---

> > ### Author Response · Authors · 2024-11-23
> > **Follow up on Reviewer 8tDw**
> >
> > We thank the reviewer for the active engagement in the discussion period and also for providing insights on how the baselines were implemented in [6].
> >
> > It seems to us that all the remaining points are related to the evaluation in [6] – the environments discussed in the response are customized environments from [6] and the baselines highlighted by the reviewer (BEN and BoostrapDQN) are for discrete domains and [6] extends them for the continuous setting (as highlighted by the reviewer). We think both, new benchmark environments and extensions of these discrete algorithms to continuous spaces, are relevant contributions to the RL community.
> > However, [6] is a concurrent ICLR submission and according to the ICLR guidelines (https://iclr.cc/Conferences/2025/FAQ) we are not required to compare our work to that paper. Excluding the concurrent work, we are already comparing to the strongest baseline in [6], REDQ, in addition to OAC as suggested by the reviewer.
> >
> > **Response to reviewer’s comment**: “Most Bayesian methods can be added on top of existing algorithms as they just affect exploration. I have added Bayesian exploration to many approaches, including PPO, DQN, RedQ etc”
> >
> > Could the reviewer please provide any published papers to validate this claim? We would be happy to include them in the related works. Moreover, we are not aware of any Bayesian RL algorithm or combination thereof being combined with and outperforming SOTA visual control methods such as DrQv2 or DrM on challenging deep RL tasks such as the humanoid and dog tasks from DMC. More importantly, we acknowledge that the reviewer has vast knowledge of Bayesian RL algorithms and we are happy to highlight the strengths of these methods mentioned by the reviewer in our related works. If the reviewer can recommend additional papers, that showcase the flexibility, and scalability to challenging visual control tasks of BRL methods, we would be happy to include them in our related works.

---

> > > ### Comment · Reviewer_8tDw · 2024-11-28
> > >
> > > `It seems to us that all the remaining points are related to the evaluation in [6]'
> > >
> > > No, I was using this as an example as what I expect for a submission in this area, mostly to rebut the point that `there was no exact implementation online in the exact area our paper is working in' is not an acceptable reason to ignore that work as it is clearly possible to implement.
> > >
> > > The purported benefit of MaxInfoRL is about exploration. To quote the authors' abstract: `In this work, we introduce a framework, MaxInfoRL, for balancing intrinsic and extrinsic exploration'. Given that Bayesian methods are designed to optimally balance exploration with exploitation, it is extraordinary to me that the authors did not include any of these methods or consider them as a Baseline.
> > >
> > > The only difference between frequentist and Bayesian methods is the former optimises true returns whereas the latter optimise the predictive returns. Any method to learn optimal policies in the standard RL setting is directly applicable to the Bayesian RL problem.  VariBAD [7] trains with PPO and is clearly compatible with any related method (RedQ, SAC...).  BBAC builds on a general actor-critic framework and any other actor critic can be used in its place [3]. [4][2] Builds on DQN . Unless the authors can formally demonstrate this is not the case (i.e. PROVE, not rely on arguments like `there doesn't exist a paper in this one specific visual control domain, therefore it is not possible to apply' ) I think this is disingenuous and I don't find this an acceptable argument.
> > >
> > > I maintain my position and think my score is correspondingly generous.
> > >
> > > [7]  Zintgraf et al. VariBAD: A Very Good Method for Bayes-Adaptive Deep RL via Meta-Learning. https://arxiv.org/abs/1910.08348

---

> > > > ### Author Response · Authors · 2024-11-28
> > > >
> > > > We thank the reviewer for additional references, we will add them in our literature review. As mentioned in the previous comment, on most tasks considered in [6] REDQ, which is already among our baselines, performs better than the Bayesian baselines (BEN and BootDQN-P). Therefore, we decided to use OAC as it complements our baselines the best and was not considered in [6].
> > > >
> > > > ``There doesn’t exist a paper in this one specific visual control domain, therefore it is not possible to apply'' -- While to the best of our knowledge, the first part of this sentence is correct (for off-policy actor-critic methods), we want to mention that we never claimed the second part, and it is not a claim we are going to include in the paper.
> > > >
> > > > Thank you for your feedback.

---

### Official Review · Reviewer_4PdE · 2024-11-02

**Soundness:** 3
**Presentation:** 4
**Contribution:** 3
**Rating:** 8
**Confidence:** 3

**Summary:**

This paper studies how to balance intrinsic and extrinsic exploration. The author gives a novel framework called MAXINFOR by combining the use of intrinsic rewards and  Boltzmann exploration, provides theoretical results within a simplified setting of stochastic multi-armed bandits, and shows that MAXINFORL outperforms existing algorithms on standard deep RL benchmarks.

**Strengths:**

1. This paper provides a novel framework for RL exploration. The idea of using intrinsic rewards and  Boltzmann exploration to balance intrinsic and extrinsic exploration is very insightful.

2. I appreciate that the author provides both theoretical and experimental results, effectively demonstrating that the proposed algorithm may actually outperform existing methods.

3. This paper is well-organized and easy to follow.

**Weaknesses:**

1. It is good to provide study problems form both theoretical and experimental perspectives, however, I think the theoretical result which only gives a convergence result seems to be weak. So is it possible to prove a regret bound for your algorithm under certain conditions?

2. I think it is good to provide more comparisons with existing algorithms.

3. I am also curious whether the method in this paper applies to tabular RL. I think it is easier to get more quantitative theoretical results (like a regret bound) within a tabular setting. Reader can more intuitively see the superiority of your method from a theoretical perspective.

**Questions:**

See Weakness for questions.

---

> ### Author Response · Authors · 2024-11-17
> **Response to Reviewer 4PdE**
>
> Thanks for your feedback on our work and also for acknowledging our algorithmic insights and experimental evaluation.
>
> ***Weaknesses***
>
> **Convergence result**: We suppose the reviewer is referring to the convergence result in Theorem 3.1. This result is inspired by the SAC paper [1], where they also show that the modified critic and policy updates give convergence. This is a crucial result as in our setting it means that our proposed policy and critic update rules give convergence. However, as also in the SAC paper, it does not take the learning aspect into account and is a consistency result for the Bellman updates.
>
> **Baselines**: As highlighted by the reviewer, the main objective of our method is to combine intrinsic and extrinsic exploration in a principled manner. To this end, we use intrinsic exploration baselines such as curiosity and disagreement for our comparison. In addition, we also compare our algorithm with [2, 3] which also try to combine extrinsic and intrinsic rewards. Furthermore, we also consider widely applied naive exploration baselines such SAC and REDQ in our experiments. Across all our experiments, we show that our approach outperforms the aforementioned methods. In addition, we show that our method also scales to complex high dimensional settings from visual control tasks, where we also compare it with SOTA (model-free) baselines: DrQ and DrQv2. All our experiments are evaluated on well-established deep RL benchmarks, including the recent HumanoidBench benchmark [4]. However, following the feedback from the reviewer, we have added additional experiments, where we compare our approach with DrM [5] on the challenging humanoid and dog tasks from the deepmind benchmark suite (Figure 18). Furthermore, we have also added another experiment comparing OAC [6] (a baseline suggested by reviewer 8tDw) to the paper (Figure 17). We hope through this we can convince the reviewer of our empirical evaluation.
>
> **Regret bound in the tabular setting**: We agree that it would be interesting to derive a regret bound for the algorithm, even in the simplified tabular setting. In fact, this is a direction we are actively pursuing for future work, where we aim to theoretically study the algorithm further. We think results from works such as [7] are a positive sign and we are hopeful of obtaining a regret bound. However, the focus of this work was mostly on the deep RL algorithm and showing the diversity and scalability of the method, therefore we decided to keep the scope of our paper focused more on those aspects.
>
> We hope our responses adequately addressed the reviewer’s concerns and would be happy if they prompt the reviewer to improve our score. Thanks once again for your insightful feedback on our work.
>
> **References**:
> 1. Haarnoja, Tuomas, et al. "Soft actor-critic algorithms and applications." arXiv preprint arXiv:1812.05905 (2018).
>
> 2. Chen, Eric, et al. "Redeeming intrinsic rewards via constrained optimization." Advances in Neural Information Processing Systems 35 (2022)
>
> 3. Burda, Yuri, et al. "Exploration by random network distillation." arXiv preprint arXiv:1810.12894 (2018).
>
> 4. Sferrazza, Carmelo, et al. "Humanoidbench: Simulated humanoid benchmark for whole-body locomotion and manipulation." arXiv preprint arXiv:2403.10506 (2024).
>
> 5. Xu, Guowei, et al. "Drm: Mastering visual reinforcement learning through dormant ratio minimization." ICLR (2024).
>
> 6. Ciosek, Kamil, et al. "Better exploration with optimistic actor critic." Advances in Neural Information Processing Systems 32 (2019).
>
> 7. Sukhija, Bhavya, et al. "Optimistic active exploration of dynamical systems." Advances in Neural Information Processing Systems 36 (2023)

---

> > ### Author Response · Authors · 2024-11-20
> > **Follow up on response to Reviewer 4PdE**
> >
> > Dear Reviewer,
> >
> > Having addressed your concerns, we would appreciate your feedback during this discussion period. Let us know if there are any further questions that we can clarify, otherwise, we would appreciate it if you would consider increasing your score.
> >
> > Thanks!

---

> > > ### Author Response · Authors · 2024-11-25
> > > **Follow up on response to Reviewer 4PdE**
> > >
> > > Dear Reviewer,
> > >
> > > As the discussion period deadline approaches, we would appreciate your feedback on our previous response. Thanks for your active engagement in the review process.

---

> > > > ### Comment · Reviewer_4PdE · 2024-11-26
> > > >
> > > > Thanks for your reply. It addresses my questions. I will adjust my score accordingly.

---

> > > > > ### Author Response · Authors · 2024-11-26
> > > > >
> > > > > We thank the reviewer for reevaluating and increasing our score. We are happy to hear that our rebuttal could address their concerns.

---

### Official Review · Reviewer_bvBz · 2024-11-04

**Soundness:** 2
**Presentation:** 2
**Contribution:** 2
**Rating:** 5
**Confidence:** 4

**Summary:**

The paper addresses the problem of exploration in reinforcement learning (RL). Whereas both randomization of actions (like ϵ-greedy and Boltzmann exploration) and information gain maximization have previously been proposed as methods for exploration in RL, the paper suggest combining both. Furthermore, it adapts a way of automatically balancing the amount of exploration that is usual in random exploration methods to information gain. The method can be combined with a wide range of off-policy RL algorithms such as SAC. The paper provides strong empirical results on several benchmarks including vision-based humanoid locomotion.

**Strengths:**

- the presentation of the paper is reasonably clear and easy to read
- the paper addresses a broadly important problem
- the idea is simple and sensible
- the empirical performance as presented appears strong
- the authors provide code for their method

**Weaknesses:**

- originality: the paper pretty much just combines the ingredients from two approaches already present in the literature: info-gain and Boltzmann exploration as present in SAC
- the weaknesses and limitations of the proposed method are not addressed in the paper
- while the paper provides a theoretical analysis in the bandit setting, the link to the full RL setting is not clearly established
- the computational overhead of maintaining and training the ensemble models is not thoroughly discussed or quantified, which seems crucial - some of the baseline methods are explicitly geared toward speed and scalability, so the paper should address this point

- page 4: the use of upper bound on information gain seems inappropriate -- you're trying to *maximize* information gain, instead shifting towards to maximizing an upper bound on information gain is just methodologically wrong. If you're crudely approximating the information gain of any distribution by the information gain of the independent multivariate normal with the same variances, you should say that you are doing that.
- p. 5, eq. (7) seems to introduce an arbitrary one-to-one trade off between information (in nats) and the reward, that can have arbitrary scale

**Questions:**

Questions:
- I would appreciate more insight into when / how the method works, and in particular why keeping the naive exploration element is still important. If you claim that info gain is more directed/principled, why not use just that? Could you provide ablations illustrating the importance of (random-action) Boltzmann exploration?
- page 4: the \epsilon-greedy method seems ill-founded: the exploratory action is taken according to the *optimal* Q-function for the intrinsic reward. However, the optimal policy is not subsequently followed (which is an assumption behind an optimal Q-function). In particular, this can result in taking an exploratory action motivated by information a purely exploratory policy could gain in several steps, while gaining no information in the first few. However, since the actual policy is likely to immediately revert to exploit behaviour, the information is never gained. In light of this, does your epsilon-greedy policy make sense?
- on p. 6, you state that the constraint "(iii) it is tight" - what does that mean?
- just to clarify, are you also planning to make the code public (e.g. on Github) upon publication?

Minor comments and suggestions:
- "c.f. is repeatedly used inappropriately as "see" - it usually means "compare to"
- I think eq. 8 would benefit from more explanation/intermediate steps
- the autotuning at the bottom of page 5 does not describe autotuning per se - as listed in this optimization, alpha would go to 0 or infinity. It becomes "autotuning" only once the optimization is done gradually, e.g. using SGD. I think it would be useful to mention that explicitly.
- p. 6 below eq. (10) - the second sentence implies that we can attain arbitrary policy entropy. This is generally not true.
- eq. (12) - since you do that for the Bellman operator, I'd suggest also making the dependence of V on the policy explicit

---

> ### Author Response · Authors · 2024-11-17
> **Response to Reviewer bvBZ**
>
> We thank the reviewer for their feedback. Particularly, we are happy that the reviewer acknowledged the simplicity of our approach and the strong empirical performance. This is a central strength we wanted to communicate with our work: well-established and simple findings from naive exploration methods can be easily integrated with intrinsic exploration objectives to achieve state-of-the-art performance. Below we address the points raised by the reviewer.
>
> ***Weaknesses***
>
> **Originality – paper combines ingredients of existing approaches**: While we understand the concern raised by the reviewer, we think this is in fact a strength of our approach. Namely, Boltzmann exploration, soft-Q learning, and automatic coefficient tuning are well-studied and widely applied in the RL community. Similarly, several works that we discuss in the paper use information gain/intrinsic rewards for unsupervised exploration in RL. However, to the best of our knowledge, we are the first to establish this link between the two, which allows us to easily combine extrinsic rewards and intrinsic exploration objectives, thus proposing algorithms that benefit from the simplicity of naive exploration schemes but enjoy principled exploration from intrinsic rewards. Compared to other works such as [1, 2] that combine extrinsic and intrinsic rewards our approach performs much better while also being easy to implement, flexible (can be combined with other RL methods as we show), and scalable.  We thoroughly evaluate our approach in the experiments as also acknowledged by the reviewer.
>
> **Connection between bandits and RL**: Thanks for your feedback. In section A of the appendix, we tried to establish this connection and also highlighted other works such as [10–13] which leverage ideas/algorithms from multi-arm bandits for RL. Particularly, [11-13] leverage and test the algorithms for RL on real-world robotic platforms. Similar to [10], the MAB setting serves as an inspiration for our algorithm and the proposed approximations for the general deep RL case.
> We are happy to receive suggestions from the reviewer on how to further clarify this connection.  Furthermore, we also want to make it clear that while we do the analysis for the MAB case, the analysis for the deep RL method would be much more challenging and we believe also out of scope for this work.
>
> **Weaknesses/Limitations/Computational cost**: We agree that learning an ensemble of NNs has additional computational costs. We highlight this in the conclusion of the paper. However, to further underline this, we have added the following table comparing the computational cost of SAC and MaxInfoSAC to the paper (see Table 1 in Appendix E).
>
> Time to train for 100k environment steps on NVIDIA GeForce RTX 2080 Ti:
> SAC (SB3): 16.96 min +/- 1.64317 min, MaxInfoSAC (SB3): 39 min +/- 1 min, SAC (JAX): 5.6 min +/- 0.2min, MaxInfoSAC (JAX): 7.3 min +/- 0.75
>
> As the number suggests MaxInfoSac requires more time to train due to the additional computational overhead. However, our JAX implementation already bridges the computational gap and is much faster. To further reduce the computational cost, we can consider cheaper methods for uncertainty quantification such as [9] or different/cheaper methods to evaluate intrinsic rewards, e.g., [2]. We have added this discussion to the paper.
>
> **Use of upper (Gaussian) bound for the information gain**: We have adapted the paper (see Section 2.4) and put more emphasis on the fact that we approximate the information gain with this upper bound. We would also like to highlight that this approximation is commonly used by several RL algorithms [3 - 6] and they show that it works well in practice. Effectively, the objective encourages exploration in areas where we have less data/high uncertainty. Particularly, [6] gives a theoretical justification for this choice of the upper bound for the model-based (active learning) setting. We will highlight this further in the paper.
>
> **Trade-off between information gain and reward**: We agree that this is generally a nontrivial trade-off due to the different scales of the objectives. However, Boltzmann exploration methods such as SAC also result in a similar trade-off and the auto-tuning approach used in these methods achieves strong performance and is widely applied. To this end, as highlighted by the reviewer, we leverage ideas from them in MaxInfoRL. Furthermore, note that we take a similar approach as in [2] of normalizing the intrinsic rewards during training, which also helps with the scaling (see section E in the appendix).

---

> > ### Author Response · Authors · 2024-11-20
> > **Follow up on our response to Reviewer bvBZ**
> >
> > Dear Reviewer,
> >
> > Having addressed your concerns, we would appreciate your feedback during this discussion period. Let us know if there are any further questions that we can clarify, otherwise, we would appreciate it if you would consider increasing your score.
> >
> > Thanks!

---

> > ### Comment · Reviewer_bvBz · 2024-11-23
> >
> > Dear Authors,
> >
> > thank you for your detailed responses to my questions. A few reactions:
> >
> > **Originality**: I see the strength side of things and don't consider this a definitive obstacle, but still a paper showing a new simple idea that works would be stronger on the originality front than one just combining two things that were previously shown to work - so still a slight weakness on the originality front.
> >
> > **Bandits**: Thank you for the clarification. I totally accept that ideas from bandits can be useful inspiration for RL approaches. Even some theoretical results from bandits have good analogues for general RL. But I feel that exploration in particular is a problem that is fundamentally harder in RL (e.g. the already discussed case of $\epsilon$-greedy would work fine for bandits, but doesn't translate into RL and you've now even confirmed it empirically that switching over multi-step periods works better). I think yes, it's good for an RL exploration method to work also in bandits, and it's nice that you show this theoretically, but I consider this limited theoretical evidence for RL in general.
> >
> > **Computational cost**: Thank you. I accept that the computational cost is acceptable.
> >
> > **Use of upper (Gaussian) bound for the information gain**: Thank you. I think I can accept this as a heuristic approximation. Just the formulation via an upper bound was misleading and I'm glad that you're changing it.
> >
> > **Importance of naive exploration**: Thank you for clarifying and adapting the figure. I think the explanation makes sense and am glad to see also an empirical comparison.
> >
> > **Clarification on $\epsilon$–greedy**: Thank you for clarifying. Yes, the switching after multiple steps seems to make sense from that perspective - the two Q-functions are valid for $\epsilon=1$ and 0 respectively. As previously mentioned, the $\epsilon$–greedy would make sense in the bandit case, but not necessarily in the RL case. Since this is not your flagship method, nor does your main method build on this, I'd probably lean in favour of removing the section on $\epsilon$–greedy.
> >
> > **Tightness of the constraint**: Thank you for clarifying.
> >
> > **Code:** I just noticed the code does not contain scripts for running most experiments. For good reproducibility, I would suggest adding both complete scripts for all experiments, and notebooks where you analyse the results. Since the main argument for your method seems empirical, I consider this point especially important. Would you be able to release that?

---

> ### Author Response · Authors · 2024-11-17
> **Response to Reviewer bvBZ continued**
>
> ***Questions***
>
> **Importance of naive exploration**: Thanks, this is a very interesting question. One reason we believe having a stochastic policy helps empirically is that it introduces diversity to the collected data if we repeat the rollouts between different episodes or across several parallel environments. Moreover, usually, actor-critics are not aggressively updated in RL to maintain stability during training. Thus, without a stochastic policy, we might have the risk of collecting very similar samples from one episode to another. Furthermore, we believe it also introduces additional stochasticity during training which could also be beneficial. Following the reviewer’s suggestion, we have adapted Figure 11 in the paper. In Figure 11, we compare a version of Maxinforl (with DrQv2) which uses a deterministic policy with the standard version (stochastic policy). We run this comparison on the challenging, high-dimensional, pixel-based exploration task: Humanoid-walk (also noted by [7] as being a hard exploration task). The figure shows that Maxinforl without noise performs better than DrQv2 with noise. However, including the noise during exploration has an additional significant gain in performance.
>
> **Clarification on $\epsilon$–greedy**: Thanks for this insightful question. We think the strategy would still make sense since the $\epsilon$ parameter is gradually decreased from 1.0 to 0.0. Therefore, during the initial stages of training the intrinsic exploration policy is sampled much more frequently effectively collecting trajectories that focus much more on intrinsic exploration. Towards the end of training, we focus more on the extrinsic reward. A similar strategy of greedy in the limit with infinite exploration (GLIE) is also used in [8]. However, we agree that perhaps a better approach would be to switch between the exploration-exploitation policies after several steps or an episode. Empirically, this is what did for our experiments in Figures 14-15, where we switched between the two strategies every 32 steps. However, to illustrate that a switch at every step also could work, we have added an additional experiment (see Figure 16) which switches between the two policies at every step. We compare this to the one where we have a switching frequency of 32 steps and show that while both approaches work, the 32-step frequency performs better.
>
> **Tightness of the constraint**: What we meant with tight is that asymptotically when the policy converges, the target policy would give similar information gain as the true policy and thus the constraint will be satisfied. However, we see how this comment can be misleading and will remove it from the paper to avoid confusion.
>
> **Code**: Yes, we will make the submitted code public.
>
> Minor comments and suggestions: Thanks a lot for pointing these out. We have already started addressing them and will adapt the paper according to the proposed suggestions. To have an interactive rebuttal period, we have already added some of the changes discussed above to the manuscript.
>
> We thank the reviewer once again for their feedback and hope our responses adequately address the raised concerns. We’d be happy if the reviewer would consider revising our score and are happy to answer any other open questions. Thanks again for your feedback and active engagement in the review process.
>
> **References**
> 1. Chen, Eric, et al. "Redeeming intrinsic rewards via constrained optimization." NeurIPS (2022)
>
> 2. Burda, Yuri, et al. "Exploration by random network distillation." arXiv preprint arXiv:1810.12894 (2018).
>
> 3. Sekar, Ramanan, et al. "Planning to explore via self-supervised world models." ICML, 2020.
>
> 4. Mendonca, Russell, et al. "Discovering and achieving goals via world models." NeurIPS (2021): 24379-24391.
>
> 5. Sancaktar, Cansu, Sebastian Blaes, and Georg Martius. "Curious exploration via structured world models yields zero-shot object manipulation." NeurIPS (2022)
>
> 6. Sukhija, Bhavya, et al. "Optimistic active exploration of dynamical systems." NeurIPS (2023)
>
> 7. Yarats, Denis, et al. "Mastering visual continuous control: Improved data-augmented reinforcement learning." ICLR (2022)
>
> 8. Parisi, Simone, Alireza Kazemipour, and Michael Bowling. "Beyond Optimism: Exploration With Partially Observable Rewards.", NeurIPS (2024)
>
> 9. Osband, Ian, et al. "Epistemic neural networks." NeurIPS (2023)
>
> 10. Cesa-Bianchi, Nicolò, et al. "Boltzmann exploration done right." NeurIPS (2017).
>
> 11. Calandra, Roberto, et al. "Bayesian optimization for learning gaits under uncertainty: An experimental comparison on a dynamic bipedal walker." Annals of Mathematics and Artificial Intelligence 76 (2016).
>
> 12. Berkenkamp, Felix, Andreas Krause, and Angela P. Schoellig. "Bayesian optimization with safety constraints: safe and automatic parameter tuning in robotics." Machine Learning 112.10 (2023)
>
> 13. Widmer, Daniel, et al. "Tuning legged locomotion controllers via safe bayesian optimization." CORL, 2023

---

> ### Author Response · Authors · 2024-11-23
> **Response to Reviewer bvBz's comments**
>
> We thank the reviewer for their feedback and active engagement in the rebuttal period. We are glad to see that our rebuttal has addressed most of the reviewer's concerns. Below is our response to the remaining points.
>
> **Originality**: As highlighted in our initial response, we think our method precisely proposes a new simple idea that connects two existing RL approaches and results in a simple, flexible, and scalable class of algorithms. Therefore, we still think our approach itself for combining and auto-tuning of intrinsic and extrinsic rewards is original and we have also not seen prior work taking a similar approach as ours.
>
> **Epsilon–greedy**: Thanks for your feedback. We would like to clarify this point further. In particular, the epsilon–greedy extension we propose is very flexible. Effectively, we can decide on a schedule for epsilon and based on the schedule, the agent will alternate between (intrinsic) exploration and exploitation actions.
> Therefore, the case where the agent switches after multiple steps is also part of our proposed formulation. In the extreme case, we can use a schedule for epsilon, which at the beginning of each episode decides to either explore or exploit and then maintains this strategy for the whole episode, i.e., $\epsilon = 1$ for the remainder of the episode. We will add this explanation as a further clarification to the paper.
> Furthermore, we would like to highlight that the comparison we made between the two different schedules for epsilon is purely empirical and we cannot draw any theoretical conclusions from it.  Moreover, we acknowledge that regret analysis for the bandit setting does not transfer directly to the RL case. However, given results from prior works [6, 8], we have confidence that for an appropriate schedule, our epsilon–greedy strategy could also result in sublinear regret for the RL case. We leave this extension for future work and will also further clarify in the paper that our analysis is for the bandit setting and extending it to the RL case is an interesting direction for future work.
>
> **Code**: We agree that providing code to reproduce our results is important and we want our code to be easily accessible for all RL researchers and practitioners. Therefore, we plan to release all our code and instructions to ensure good reproducibility. We are working on it already and we will have the release ready together with the camera-ready version.
>
>
> We hope our response addresses the reviewer’s remaining concerns and given that we could address all the other concerns raised in the review, we would appreciate it if the reviewer would consider increasing our score and are happy to answer any other open questions. Thanks again for your active engagement.

---

> > ### Author Response · Authors · 2024-11-25
> > **Follow up on response to reviewer bvBz**
> >
> > Dear Reviewer,
> >
> > As the discussion period deadline approaches, we would appreciate your feedback on our previous response.
> > Thanks for your active engagement in the review process.

---

> > > ### Author Response · Authors · 2024-11-29
> > >
> > > Dear Reviewer,
> > >
> > > Having addressed your concerns, we would appreciate your feedback during this discussion period. Let us know if there are any further questions that we can clarify, otherwise, we would appreciate it if you would consider increasing your score.
> > >
> > > Thanks!

---

> > > > ### Comment · Reviewer_bvBz · 2024-12-03
> > > >
> > > > Thank you for the responses.
> > > >
> > > > Regarding $\epsilon$-greedy, I accept the version switching after k steps. However, in general, I'd be careful mixing multiple similar contributions in a paper, unless their comparison is an important aspect of the paper. However, it's not a major issue to me.
> > > >
> > > > I don't have any further questions at this point - once more thank you for all your responses and engagement!

---

> ### Author Response · Authors · 2024-12-03
>
> Dear Reviewer,
>
> Thanks for your response. We are glad we could address the remaining open concerns. Since you do not have any further questions, we would be very happy if you would consider increasing our score.
>
> Thanks again for your active engagement in the rebuttal period.

---

### Public Comment · ~Alessio_Russo1 · 2025-03-13
**Missing related work on exploration techniques**

Dear authors, in the related work sections you missed some key recent works, which are related to active exploration and best policy identification in RL.

- Russo, Alessio, and Alexandre Proutiere. "Model-free active exploration in reinforcement learning." Advances in Neural Information Processing Systems 36 (2023): 54740-54753.
- Russo, Alessio, and Filippo Vannella. "Multi-reward best policy identification." Advances in Neural Information Processing Systems 37 (2024): 105583-105662.

I would appreciate if you could cite these works as these are relevant information-theoretical based techniques, also applied in DeepRL.

---

### Meta-Review · Area_Chair_jD1X · 2024-12-22

**Metareview:**

This paper studies a new exploration strategy that automatically balances between global exploration (e.g. UCB, the so-called intrinsic exploration) with local exploration (e.g. $\epsilon$-greedy, the so-called extrinsic exploration). Empirical benefits are demonstrated on visual control tasks. Theoretically, the paper present a regret bound on the multi-armed bandits problem. The reviewers are unanimously recommending acceptance.

**Additional Comments On Reviewer Discussion:**

NA

---

### Decision · Program_Chairs · 2025-01-22

Accept (Poster)